

# Assessment of possible volcanic hazards in Germany with regard to repository site selection

Ulrich Schreiber[1], Gerhard Jentzsch[2]

[1]Department of Geology, University of Duisburg-Essen, Essen, 45141, Germany, retd.
[2]Institute for Geosciences, University of Jena, Burgweg 11, 07749 Jena, Germany, retd.

*Correspondence to*: Ulrich Schreiber (ulrich.schreiber@uni-due.de) and Gerhard Jentzsch (gerhard.jentzsch@uni-jena.de)

**Abstract.**

In the context of selecting a site for the long-term disposal of radioactive, heat-generating waste in deep geological formations, the potential for future volcanic activity within the next 1 million years must be systematically evaluated. This assessment

draws upon an integrated analysis of geological, geochemical, and geophysical datasets, as well as isotopic measurements of crustal and mantle-derived gases. Relevant data sources include teleseismic imaging, long-term seismic and microseismic monitoring—particularly deep earthquake patterns—and geodetic observations of vertical crustal movements. Additional insights are provided by geological and mineralogical studies that inform the spatial distribution and petrogenesis of volcanic rocks. When combined with geophysically derived mantle anomalies and radioisotopic age data for volcanic centers, these

datasets enable the delineation of areas with an elevated probability of future volcanism. Special focus is given to the Quaternary volcanic provinces of the Eifel and Vogtland, which are identified as regions with a significantly increased likelihood of renewed activity. The outermost volcanic centers in these regions are used to define preliminary hazard perimeters. A conservative safety buffer of 25 km beyond these limits is adopted to define the exclusion zone boundary for deep geological repositories. In the Vogtland region, known for its characteristic earthquake swarms, seismic epicenters are

equated with volcanic centers to delineate zones of potential recurrence. The extent of this area is adjusted accordingly based on seismic swarm distribution and geophysical data. A major secondary hazard associated with volcanism in the Eifel is the potential damming of the Rhine River within its narrow Middle Rhine Valley by lava flows or tephra deposits. Prolonged blockage of the river would result in extensive upstream flooding, affecting the Upper Rhine Graben and its tributary valleys. Two regions in Germany—north of the Westerwald and east of the Black Forest—are classified as having a low probability

of future volcanism within the next 1 million years. In the Tertiary volcanic fields, no volcanic activity is expected within the next 1 million years due to their advanced age and normal mantle and gas compositions.

## 1 Introduction

In the context of selecting a deep geological repository for heat-generating radioactive waste, the Working Group on Site Selection Procedures (AkEnd, 2002) determined that potential future volcanic activity must be considered when evaluating

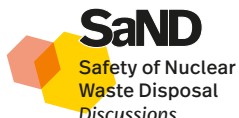

sites in Germany. A comprehensive report—forming the basis of this review—defined exclusion zones where renewed
volcanic activity may occur within the next one million years, with varying probabilities (Quote from the BGE report, not yet
available). These zones were incorporated into the BGE's interim report on sub-areas (updated 3 March 2022) and underpin
the exclusion criteria outlined in Section 22, Chapter 5 of the Site Selection Act (StandAG, 2017/2019). The present work
builds on the 2022 geological dataset and summarizes the methodology used to delineate these zones. According to regulatory
specifications, areas identified as being at risk must be excluded from consideration, along with an additional 10 km safety
buffer. The assessment time frame extends 10 million years into the past and one million years into the future. The assessment
of volcanic hazard focuses on potential eruptions and their possible effects on a planned repository for radioactive waste.
Several influencing factors depend on the type of volcanism and must be considered separately for the operational phase and
the post-operational (monitoring) phase. The operational phase encompasses the construction and emplacement of waste within
the repository. Once disposal is complete, the facility is sealed, initiating the post-operational phase with long-term monitoring.
The retrieval of radioactive waste is to remain possible for up to 500 years following the operational phase (Site Selection Act,
2017/2023). Surface effects of volcanic activity may include regional uplift affecting repository seals, volcanic-tectonic
earthquakes, forest fires triggered by eruptions, and maar formation. Uplift could impact an area of approximately 500–
1,500 km²; forest fires may extend across 50–200 km²; and maars may form over similarly sized regions. The area within
approximately 5 km of an eruption center would be particularly threatened by highly explosive magma–water interactions and
associated shock waves. Lava flows and lahars could inundate valleys and affect broader areas over extended periods due to
backflow processes. Additionally, widespread ash deposition of varying thickness may occur, potentially obstructing surface
infrastructure and limiting access to underground facilities. Although direct magma intrusion into a repository is considered
highly improbable, subsurface impacts must still be accounted for. These include thermal stress, volcanic earthquakes, and
induced fault movements, any of which could compromise the integrity of the repository and reduce the effectiveness of
engineered and geological barrier systems by facilitating groundwater ingress. Ultimately, any scenario that could lead to the
release of radioactive materials must be entirely prevented. Particular attention must be paid to Plinian eruptions (e.g., Laacher
See-type events) of highly evolved magmas, which can result in extensive ash fallout over large regions. Assessing the
associated hazard is challenging due to the broad spatial extent. A key issue is determining the critical tephra thickness that
constitutes an unacceptable risk to a repository. In this context, the different timescales of the operational phase (several
decades), the post-retrieval phase (up to 500 years), and the long-term safety period (one million years) must be evaluated
separately.

## 1.1 Criteria for Potential Volcanic Hazard

The assessment of future volcanic hazards is based on multiple criteria, including: teleseismic studies of mantle temperature
anomalies (e.g., mantle plumes), isotopic analyses of gases from mofettes (e.g., helium; $CO_2$ considered to a limited extent),
tectonic characteristics of the region, including recent seismic activity, age and spatial distribution of volcanic centers and
tephra deposits, and geodetic studies of relative vertical crustal movement. The potential hazard posed by a site depends on

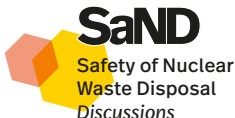

whether the volcanism is likely to be explosive or purely effusive. The expected magnitude of future eruptions can be estimated by examining volumes of past eruptive products. To accommodate worst-case scenarios, the upper bound of this range must

be considered. This enables approximations of the potentially affected area and, by integrating the likely eruption centers with regional topography, allows the prediction of lava flow paths, pyroclastic density currents, and possible lahars. The height and distribution of tephra deposits are strongly influenced by prevailing wind conditions. Therefore, safety buffers around repository sites must be sufficiently large to accommodate even low-probability eruption scenarios. This requires a clear definition of potential volcanic impacts and an assessment of which effects are tolerable over different timeframes.

**2 Overview of mantle structure beneath Central Europe**

To assess the potential for future volcanic activity in Germany over the next million years, results from multiple teleseismic studies investigating the structure of the European mantle have been evaluated. These studies, conducted over the past three decades by various research groups (e.g., Goes et al., 1999; Ritter et al., 2001; Keyser et al., 2002; Walker et al., 2005; Koulakov et al., 2009; Zhu et al., 2012; Handy et al., 2021), employed diverse seismic methodologies to analyze anomalies in

the travel times of P- and S-waves, thereby detecting large-scale variations related to temperature and/or fluid content in the mantle. Goes et al. (1999) identified an anomalously hot mantle beneath the Rhenish Massif and parts of southern Germany, particularly at the base of the lithosphere, approximately 100 km deep (Figure S1, Supplement). Comparison with the Massif Central, which is also characterized by Quaternary volcanism, revealed a similar mantle anomaly. A diffuse low-velocity zone was found to extend from the Eifel region through the Vogelsberg towards the Bohemian Massif. Importantly, the study found

no evidence of this anomaly extending into the lower mantle beyond a depth of 600 km. Koulakov et al. (2009) analyzed variations in P- and S-wave velocities across the mantle beneath Central and Southern Europe down to depths exceeding 600 km. Their tomographic profiles revealed mantle anomalies associated with known volcanic provinces, including the Massif Central, the Eifel, Vogtland, and southern Germany. Based on travel-time anomalies (see profiles in Figure S2, Supplement), they inferred thermally or compositionally anomalous regions beneath these volcanic fields. Profile 12 (Figure S3,

Supplement), oriented NNW–SSE, traverses western Germany, intersecting both the Rhenish Massif and the area west of the Upper Rhine Graben. P-wave velocities are significantly reduced over a lateral extent of more than 500 km and to depths exceeding 200 km, with the strongest anomalies located beneath the Eifel within the lithospheric mantle. S-wave data show a narrow anomaly extending from the asthenosphere up to the lithosphere–crust boundary. This anomaly broadens at greater depth, with an even more pronounced reduction in S-wave velocities. Outside the Eifel region, the upper mantle appears

seismically normal, except for a narrow low-velocity zone beneath the Upper Rhine Graben, which seems to connect to the deeper Eifel anomaly. In the Vogtland region, profile 8 (Figure S4, Supplement) reveals reduced P-wave velocities over a broader zone that extends into the crust. Below this, a low-velocity region in the asthenospheric mantle extends northeastward toward the Baltic Sea. While no S-wave anomalies are observed in the crust above Vogtland, significant velocity reductions occur in both the lithospheric and asthenospheric mantle directly beneath the region. Profile 10 (Figure S5, Supplement)



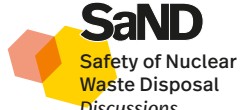

highlights mantle velocity anomalies in western and southern Germany. In addition to the anomalies beneath the Eifel (as seen in profile 12), reductions in P-wave velocities are also detected in the Urach region of southern Germany. This anomaly extends vertically from the Moho down to 200 km, and further southeast into deeper parts of the lithospheric and upper asthenospheric mantle. Reductions in S-wave velocity are evident beneath the Eifel, Hegau, and Urach; in the latter, the anomaly appears to be directly connected to the asthenosphere and extends to depths of approximately 400 km. Handy et al. (2023) argue that the

negative Vp anomalies observed at the base of the lithosphere in this region are structural in origin, rather than the result of thermal anomalies related to a hot asthenosphere. They interpret these features as remnants of thick, ancient European lithosphere, potentially dating back to the Variscan or even pre-Variscan orogenies. Zhu et al. (2012) examined mantle structure across Europe using vertically polarized shear-wave velocity anomalies (βv) at multiple depth slices (75, 175, 275, 475, and 625 km). Their analysis confirmed a strong negative anomaly beneath the Eifel in the lithospheric mantle, with a

weaker anomaly beneath the Upper Rhine Graben. Comparable anomalies were also observed beneath the Vogtland region and the Bohemian Massif. The Eifel anomaly persists at greater depths and expands at ~275 km depth to encompass large parts of southern Germany, reaching the Czech border. In contrast, the anomaly beneath the Upper Rhine Graben is no longer detectable beyond 175 km. At deeper levels (475–625 km), all velocity anomalies vanish, suggesting no direct connection to the lower mantle.


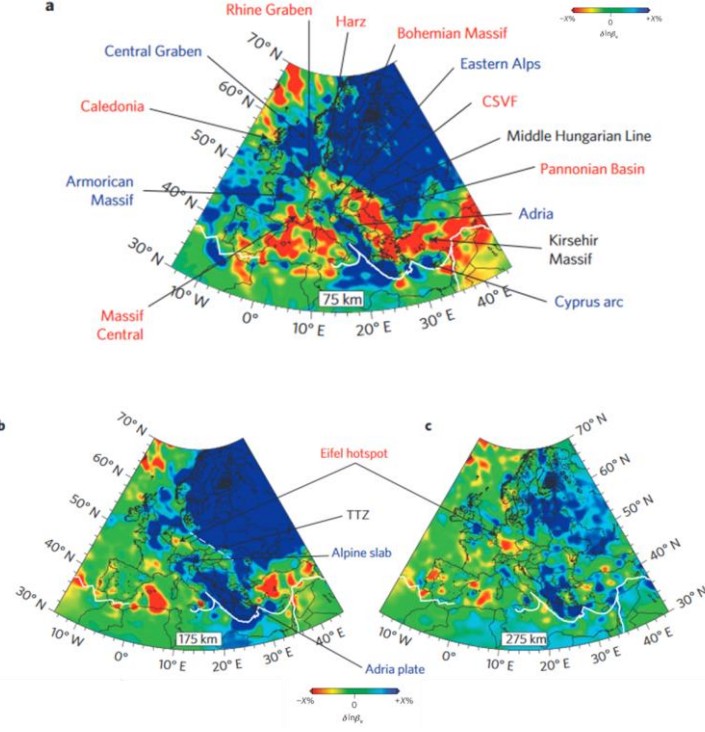



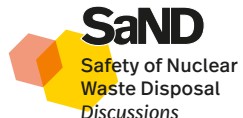

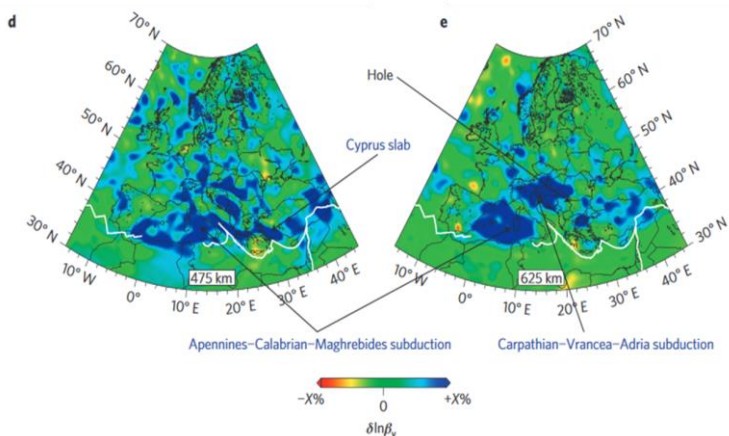

**Figure 1 from Zhu et al. (2012): Horizontal cross-sections of relative perturbations of vertically polarized shear waves βv, δlnβv, for model EU30 at the depth of a: 75 km, b: 175 km, c: 275 km, d: 475 km, e: 625 km. The colours indicate the relative wave-speed perturbations in relation to the 1D background model STW105 (Kustowski et al. 2008). The deviations range from -X% to + X%, with a: X = 4 and b - e: X = 3. CSVF, Central Slovakian Volcanic Field; TTZ, Tornquist-Teisseyre Zone.**

## 3 Isotopy of mofette gases in Germany and adjacent areas

In addition to seismological data, isotope analyses of gases emanating from the lithosphere provide valuable insights into mantle-derived processes that may contribute to magma generation. Most of the available isotope data from the Cenozoic volcanic fields pertain to helium, which, as a chemically inert noble gas, is particularly well-suited for geochemical fingerprinting and comparative studies. Helium naturally occurs in two isotopes, $^3$He and $^4$He, whose ratios—expressed as R-values—vary by several orders of magnitude among atmospheric, crustal, and mantle reservoirs. The ratio of a sample's helium

isotope composition to that of the atmosphere (Ra) typically ranges from 0.05 to more than 6 in Central Europe, with elevated values indicating a significant contribution of mantle-derived $^3$He. A map compiled by May (2019; Figure S6, Supplement) illustrates the distribution of elevated R/Ra values in $CO_2$-rich springs across the Rhenish Massif and surrounding regions. Notably, high R/Ra values are concentrated near Quaternary volcanic fields, underscoring an ongoing influence of mantle volatiles. Table S1 (Supplement), based on data from Grieshaber et al. (1992), lists R/Ra values measured in gases from the

East Eifel, the southeastern margin of the Taunus, the Vogelsberg region, and the southern Upper Rhine Graben. The highest values are observed in the central East Eifel, particularly in the vicinity of the Laacher See, Wehr, and Rieden calderas. The peak value of 5.87 R/Ra was recorded at Glees, indicating a substantial mantle helium component. Bräuer et al. (2018) reported similarly high Ra values in gases from the Vogtland region. In the Cheb Basin, values were comparable to those observed in the Eifel. In contrast, the Mariánské Lázně gas field exhibited generally lower R/Ra values, with a maximum of 4.6 R/Ra

measured at Prameny (Table S2, Supplement). Ufrecht (2006) extended the dataset by incorporating helium isotope analyses from the Stuttgart region. These measurements, which build upon earlier work by Grieshaber (1992), revealed R/Ra values

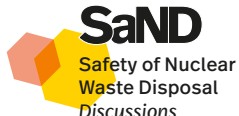

ranging between 0.05 and 0.36 (Figure S7, Supplement), indicative of predominantly crustal helium with only minor mantle input.

## 4 The Quaternary Eifel volcanic region

### 4.1 The Eifel plume

Compared to other regions of Germany, the Rhenish Massif exhibits a distinct mantle structure, which, similar to the Massif Central in France, is interpreted by most researchers as evidence for an underlying mantle plume. Ritter et al. (2001) reported a reduction of up to 3% in P-wave velocities beneath the Rhenish Massif, which they attributed to a mantle plume within the asthenospheric mantle. This plume is believed to ascend into the overlying lithosphere (Figure S8, Supplement). As the
observed seismic anomalies cannot be plausibly explained by variations in rock composition, the authors concluded that a temperature anomaly of 150–200 °C (±100 °C) is the most likely cause. Based on their findings and tomographic modeling, Ritter et al. proposed that even small volcanic fields could result from large-scale ascent of anomalously hot mantle material. They emphasized that melt generation within the plume depends on several factors, including the strength and velocity of mantle upwelling, the presence of volatiles, the thickness of the lithosphere, and the temperature contrast between the plume
and surrounding mantle. Among these, the latter two parameters are considered critical for determining the degree of partial melting. These insights also help explain the comparatively low volcanic activity observed in the Vogtland region. In evaluating the potential for future volcanism, it is essential to consider the contrast with Tertiary volcanic provinces such as the Westerwald, Vogelsberg, and Rhön, which were significantly more voluminous and active for at least ten times the duration of current Quaternary fields. If the existence of a mantle plume is confirmed, the upwelling process is likely ongoing. Ritter
(2005) estimated ascent rates of up to 6 cm per year, which suggests that significant volcanic activity may persist over the next one million years. Keyser et al. (2002) modeled shear-wave travel time anomalies associated with the Eifel plume and presented two vertical profile sections extending to depths of 500 km (Figure S9, Supplement). In both NW–SE and W–E cross sections, a prominent mantle plume anomaly is visible, extending from the lithospheric mantle down to ~180 km into the asthenosphere. Below this depth, a ~70 km gap with normal seismic velocities is observed, followed by a renewed low-velocity
zone extending to depths beyond 500 km. This two-layer structure points to significant differences in how S- and P-waves interact with mantle materials. Notably, towards the southeast (Stuttgart region), seismic velocities increase substantially. In contrast, the W–E profile shows a weaker anomaly extending eastward into the region of the Tertiary Vogelsberg volcanic field. Building on these findings, Walker et al. (2005) proposed a more nuanced model for the Eifel plume, suggesting a parabolic upwelling path from the asthenosphere into the lithosphere. According to their model, the plume's center lies between
the West and East Eifel Quaternary volcanic fields and extends westward into the Ardennes, northwestward and northward into the Lower Rhine Embayment, and eastward across the southwestern Westerwald into the Taunus region (Figure 2). To the south and southwest, parts of the Hunsrück and the Trier Basin are also affected. The model also incorporates electrical conductivity data from the mantle. As hot mantle rock ascends, the flow aligns mineral structures, resulting in anisotropic



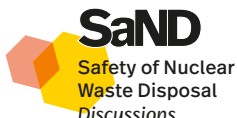

conductivity. In the case of the Eifel plume, the highest conductivity is oriented in an east–west direction, which corresponds

closely to the absolute westward motion of the European Plate. This motion forms part of the broader southwestward tectonic

drift of Central Europe at a rate of $1.9 \pm 1.4$ cm/year (Gripp and Gordon, 2002; Figure S10, Supplement).

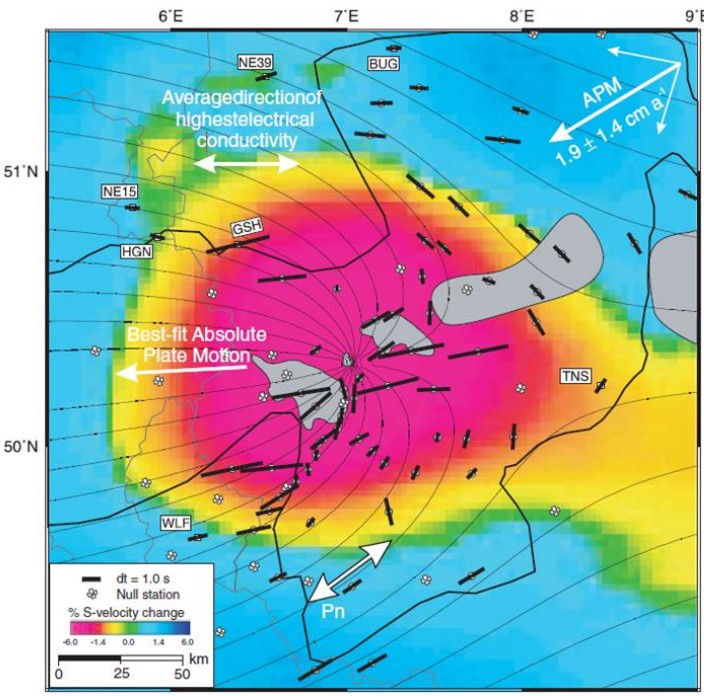

**Figure 2 from Walker et al. (2005): Summary of the results of the investigations on the Eifel plume. In addition to the model of a parabolic ascent of the asthenosphere (depth 30–100 km; shear wave anomaly according to Keyser et al. 2002), the direction of the**
**highest electrical conductivity in the mantle and the best matching absolute plate movement are shown. The determined plate movement to the west is sub-parallel to the average highest electrical conductivity in the asthenosphere (Leibecker et al. 2002). It lies within a 95% confidence interval of the absolute plate motion direction for Central Europe determined by Gripp and Gordon (2002).**

To evaluate the likelihood of future volcanic activity over the next one million years, it is essential to assess whether

lithospheric plate motion could substantially alter magmatic conditions in the underlying mantle. According to Gripp and

Gordon (2002), the current plate motion rate of $1.9 \pm 1.4$ cm/year corresponds to a displacement of approximately 5 to 33 km

per million years. These values are particularly relevant when considering the relative motion of the lithosphere over a fixed

asthenospheric plume. When a mantle plume intrudes into the lithosphere, thermal erosion causes the asthenosphere–

lithosphere boundary to rise toward the crust. In the case of the Eifel region, this boundary has been uplifted to a depth of

approximately 50 km. However, the plume conduit is entirely encased within lithospheric mantle, which thickens toward its

periphery. As the overlying lithosphere migrates laterally, the upper portion of the plume is displaced accordingly. Over

geologic timescales, this process leads to plume deflection and eventual structural offset between the lower, ascending plume

and its upper lithospheric expression. If this offset becomes sufficiently pronounced, the ascending plume may penetrate a new

segment of the lithosphere, forming a separate intrusion. The onset of such a process depends on the ascent rate of the

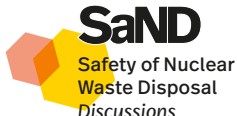

asthenospheric component of the plume, which Ritter (2005) estimated at up to 6 cm/year for a temperature contrast of 100–150 °C relative to ambient mantle. This translates to a vertical displacement of up to 60 km per million years. Given that the plume's ascent rate exceeds the lithospheric drift rate, the formation of a new volcanic field within the next million years is considered unlikely. Instead, only minor lateral shifts in volcanic activity within the established Quaternary fields are anticipated. A more recent synthesis of Eifel plume data by Ritter (2007) yielded the following conclusions: The Eifel plume

anomaly initiates at a depth of approximately 50–60 km and extends to around 410 km (Figure S11, Supplement), with an estimated diameter of 100–120 km. The observed reductions in both P- and S-wave velocities are interpreted as a result of a 100–150 °C temperature anomaly relative to the surrounding mantle. Additionally, the depth range of 50–100 km may contain a partial melt fraction of ~1%. However, to account for observed surface volcanism in the Eifel, higher degrees of melting—exceeding 5%—are required. Potential mechanisms include decompression melting during plume ascent and volatile release

from hydrous minerals. Notably, neither Ritter (2007) nor Mathar et al. (2006) observed significant evidence for plume head expansion in their surface wave analyses. This contrasts with the model proposed by Walker et al. (2005), who identified a broader asthenospheric low-velocity anomaly beneath the Rhenish Massif. The mantle plume hypothesis has recently been questioned by Dahm et al. (2020), who suggest that Quaternary volcanism in the Eifel may instead result from interactions between upper mantle upwelling and regional tectonic stress fields. In their interpretation, pre-existing lithospheric

structures—such as fossil sutures from past subduction zones and crustal discontinuities inherited from the Variscan orogeny—play a key role in guiding magmatic ascent. According to their model, magmatism is fed by reservoirs located at the crust–mantle boundary and within the lower crust. Recent studies also support the presence of so-called mush reservoirs—cold, crystal-rich, low-melt bodies—as potential sources of Quaternary volcanism (Annen, 2011). In this context, Hensch et al. (2019) identified deep low-frequency (DLF) earthquakes with magnitudes below M 2, occurring along a narrow conduit-like

zone extending from 43 km to 8 km beneath the Laacher See volcano. These events are interpreted as indicators of $CO_2$-rich fluid migration, possibly accompanied by minor magma movement, within a deep crustal or upper mantle conduit.

## 4.2 Tectonics of the West and East Eifel

The tectonic evolution of the Rhenish Massif over the last one million years has been significantly influenced by regional uplift processes. However, the temporal and spatial consistency of this uplift remains uncertain. It is not yet clear whether the

uplift has occurred as a continuous, steady-state process or as a series of episodic pulses with variable local amplitudes. Based on the elevation levels of fluvial terrace sequences, a temporal correlation has been proposed between the initiation of regional uplift and the onset of volcanic activity in the Eifel approximately 700,000 to 800,000 years ago. Estimates of total uplift since that time vary between ~140 m (Demoulin and Hallot, 2009) and ~250 m (Van Balen et al., 2000; Meyer and Stets, 2002), corresponding to average long-term uplift rates of approximately 0.1 to 0.3 mm/year. In contrast, more recent geodetic analyses

by Kreemer et al. (2020), based on continuous GPS measurements, indicate significantly higher current uplift rates of up to ~1 mm/year (Figure 3). These more rapid rates have been interpreted as evidence for an active contribution from the underlying Eifel mantle plume and its associated dynamic upwelling processes. However, these GPS-based uplift rates represent short-

term geodetic snapshots over decadal timescales and should not be directly extrapolated to infer long-term tectonic trends. More reliable projections of future uplift over geological timescales require averaging across extended temporal windows to

account for episodic variability and tectono-magmatic feedbacks.

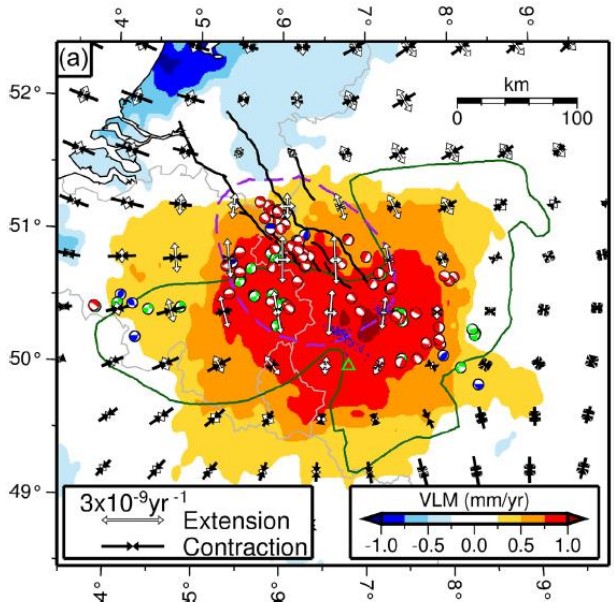

**Figure 3 from Kreemer et al. (2020): Vertical movement rates of the crust in the Rhenish Massif from GPS data, corrected for the glacial-isostatic rebound. Maximum uplift with up to 1 mm/a is shown in red. The main axes of horizontal extension and contraction are shown. The area with the greatest extension is outlined with a purple dashed line. Fault plane solutions according to Hinzen**

**(2003) and Camelbeeck et al. (2007) are given. Normal faults are in red, thrusts are in blue and strike-slip displacement is given in green. Blue dots correspond to centers of Quaternary volcanic activity. The green triangle corresponds to the center of the Eifel plume according to Ritter et al. (2001).**

A striking observation is the spatial offset between the proposed center of the Eifel plume—identified by Ritter et al. (2001) and marked by a green triangle in Figure 3—and the zone of maximum surface uplift. In contrast, the broader plume boundaries

delineated by Walker et al. (2005) exhibit a better spatial correlation with observed uplift patterns (see Figure 2). The greatest crustal extension in the region occurs within the northwestern sector, where it only partially overlaps with the West Eifel volcanic field. Notably, this area lacks significant occurrences of highly evolved volcanic rocks, such as phonolites, which may suggest that intensified uplift in this region is a relatively recent development. Conversely, the East Eifel volcanic field, which exhibits lower levels of crustal extension, is characterized by the presence of phonolitic lava domes and caldera-forming

volcanic systems. This suggests that the development of evolved magmas in the East Eifel may be influenced by local fault systems—particularly en echelon strike-slip faults with associated pull-apart or transtensional geometries. These tectonic structures can promote magma accumulation, differentiation, and explosive volcanism. The spatial mismatch between the plume center (as defined by Ritter et al., 2001), the uplift maximum, and the zones of localized extension and evolved magmatism introduces uncertainty into current interpretations of magma generation and evolution within the mantle beneath

the Rhenish Massif. If the present uplift rates were to continue uniformly over the next one million years, the central region



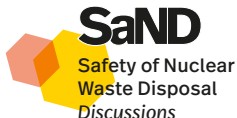

could experience a cumulative elevation gain of approximately 1000 m. The current tectonic regime of the Eifel and surrounding regions is primarily governed by the orientation and magnitude of the prevailing regional stress field (Figure S12, Supplement). Since the Miocene, the maximum horizontal compressive stress (SHmax) across Central Europe has been oriented NW–SE (Grünthal and Stromeyer, 1994).

**4.3 Quaternary volcanism of the Eifel**

The most recent volcanic eruption in the Eifel occurred approximately 11,000 years ago at the Ulmen Maar (Brauer et al., 1999). As a result, no historical records of volcanic activity exist for this region. In the central Eifel, remnants of Eocene volcanic activity—referred to as the High Eifel—are preserved and bordered to the west and east by the Quaternary volcanic fields of the West and East Eifel, respectively. For the purpose of assessing future volcanic hazard, only the Quaternary

volcanic fields are considered relevant, as their activity overprints any older tectono-magmatic influences. Quaternary volcanism began in the West Eifel around 700 ka and in the East Eifel around 500 ka (Bogaard, 1995; Schmincke, 2009; Förster and Sirocko, 2016; Tables S3, S4, Supplement). The West Eifel contains approximately 256 volcanic centers, while the East Eifel hosts around 100. In both regions, volcanic activity is concentrated in core areas, with only a few isolated centers located farther afield. In the adjacent Westerwald region to the east, Lippolt and Todt (1978) described Quaternary volcanic

rocks, though the ages of these occurrences remain uncertain. Consequently, hazard assessments for future Eifel volcanism are typically conducted under two scenarios—one that includes and one that excludes the Westerwald as a contributing volcanic area. Approximately 2,000 years prior to the Ulmen Maar eruption, the Laacher See volcano in the East Eifel erupted in a major Plinian event (Table S3, Supplement). The Laacher See tephra is exceptionally well preserved, and its distribution and thickness serve as a key reference for reconstructing the eruptive history of evolved magmatic systems in the region and for

assessing potential future impacts. Across both volcanic fields, eruption ages generally decrease from northwest to southeast. Mertz et al. (2015) identified two distinct age groups in the West Eifel, separated by a volcanic hiatus of approximately 400 ka. The older phase (720–480 ka) is located northwest of Üdersdorf and Gillenfeld, while lava flows southeast of Üdersdorf typically post-date 80 ka. The current geochronological dataset includes 39 localities, confirming this spatial–temporal separation. However, the available dates are largely from lava flows, which likely represent the final eruptive stages at each

site. Earlier eruptive activity, particularly phreatomagmatic maar formation, may substantially predate these ages. One notable outlier—a lava flow near Gerolstein—has been dated at $32 \pm 13$ ka but is located in the older, northwestern sector. Currently, only about 15% of the volcanoes in the West Eifel have been dated using modern techniques. While the southeastward migration trend of volcanism is evident, the dataset remains insufficient for establishing a robust model of mantle plume migration or for reliably estimating its velocity, as proposed by Mertz et al. (2015). This limitation is compounded by the small

number of dated samples and the presence of anomalous ages. An estimated migration rate of volcanic activity on the order of 4–5 cm/year has been derived from radiometric dating. However, this rate is not attributed to lithosphere-asthenosphere displacement. Instead, Mertz et al. (2015) emphasize a correlation with the 400 ka volcanic hiatus in the West Eifel, which coincides with the onset of activity in the East Eifel around 480 ka. During the Late Pleistocene (<100 ka), volcanic activity

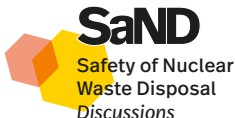

appears to have shifted back to the West Eifel, particularly in its southeastern sector, while the East Eifel remained largely
dormant—with the exception of the Laacher See eruption at ~13 ka (Schmincke, 2007). These interpretations should be treated
with caution due to the limited number of precisely dated volcanoes in both fields. For example, the Hohe Buche volcano near
Namedy in the East Eifel erupted around 100–80 ka, forming a prominent scoria cone. Its final eruptive phase produced a lava
flow that reached the Rhine River, forming a lava delta on the Younger Middle Terrace. Förster et al. (2019) propose that a
basanitic/tephritic eruption at the current site of the Laacher See occurred 24.3 ka ago, producing the Eltville Tephra. If future
geochronological studies identify additional, previously unrecognized eruptions in either field, apparent temporal gaps in
volcanic activity may be revised. Consequently, the current model of a southeastward-migrating mantle anomaly at a rate of
4–5 cm/year warrants reassessment.

## 4.4 Volcanoes of the West Eifel

While the maars of the West Eifel represent the most well-known volcanic landforms of the region, the dominant volcanic
structures are scoria cones produced by Strombolian eruptions. In approximately half of these cases, the final stages of activity
yielded sufficient magma volumes to generate lava flows. Over the ~700,000-year eruptive history of the West Eifel volcanic
field, the average recurrence interval of eruptions has been estimated at approximately 2,875 years (Shaw, 2004). However,
this average should not be interpreted as indicative of a uniform temporal distribution, as volcanic activity is likely to have
occurred in episodic clusters. Shaw (2004) also estimated magma ascent rates for three West Eifel volcanoes containing mantle
xenoliths, finding values ranging from 3 to 15 km/h. These ascent rates correspond to relatively late stages of magma
transport—after the formation of an open, continuous conduit system. They provide initial constraints on the potential
timescales involved in future eruption scenarios. Prior to the development of an open conduit, the initial ascent of magma must
exploit pre-existing zones of weakness within the crust—typically along tectonic fault planes where shear stress is minimized.
As magma rises, it displaces a leading front of fluids, composed of supercritical phases of water, carbon dioxide, and nitrogen
(scH$_2$O, scCO$_2$, scN$_2$), originating from both mantle and crustal sources. During ascent, the overpressure at the top of the
magma column relative to the surrounding rock can sustain the supercritical state of these fluids up to very shallow depths. A
sudden pressure drop occurs only when the final tens to hundreds of meters of the overlying crust are breached, causing an
abrupt phase transition from the supercritical to gaseous and vapor states. This rapid decompression is critical for explosive
eruption dynamics. As Stober and Bucher (1999) demonstrated, the energy stored in supercritical fluids can be 10 to 20 times
greater than that in an equivalent volume of superheated vapor. This disparity explains the extreme explosive power observed
in the initial phases of many maar-forming eruptions in the region.

## 4.5 The East Eifel volcanoes

The majority of volcanic centers in the East Eifel are scoria cones, many of which exhibit short lava flows or welded scoria
deposits. Similar to the West Eifel, the East Eifel includes an older volcanic region in the northwest (Schmincke, 2007). A key
distinction, however, is the presence of several large caldera complexes—namely, the Rieden volcanic system, the Wehr



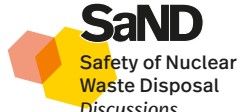

caldera, and the Laacher See volcano (LSV). These structures are interpreted as collapse calderas resulting from the evacuation of substantial magma chambers. In comparison to mafic scoria cones and lava flows, such caldera systems represent a markedly different scale and type of volcanic hazard. The Laacher See volcano is the youngest and most well-preserved volcanic system in the East Eifel, having last erupted approximately 13,000 years ago (Bogaard, 1995; Schmincke, 2007). This has enabled

detailed reconstruction of the eruption dynamics, which serve as an important analog for assessing the potential impact of similar events on a deep geological repository. Initial estimates placed the volume of erupted magma from the LSV at 5.3 km³, later revised to over 6.3 km³, in addition to 0.5 km³ of overlying country rock incorporated during the eruption (Wörner et al., 1985; Bogaard and Schmincke, 1985; Harms et al., 2004; Schmincke, 2007). As the phonolitic melt erupted, it expanded into pumice, approximately tripling in volume. Based on tephra thicknesses in both proximal and distal deposits, the total erupted

volume has been estimated at ~18 km³, from which a dense rock equivalent (DRE) of 6.3–6.5 km³ has been calculated. U–Th dating of intrusive carbonatites by Schmitt et al. (2010) suggests that the Laacher See magma system was active and accumulating for at least 20,000 years prior to eruption. Given that differentiation and cooling occur concurrently with magma emplacement, the entire chamber evolution process likely spanned several tens of thousands of years (see also Annen, 2011). A fundamental problem remains unresolved: geophysical and petrological estimates place the depth of the Laacher See magma

chamber between 4.6 and 7.8 km (Harms et al., 2004), which situates it within the brittle upper crust. The formation and expansion of such a magma chamber would require substantial crustal displacement along tectonic faults. Assuming a chamber height of ~3 km and a minimum volume of 6.3 km³, the chamber footprint would measure at least 1.5 × 1.5 km. Based on the estimated parental basanitic magma volume of 16.6 km³ (Bogaard & Schmincke, 1984), substantial additional volume remained stored in mid-crustal levels. To accommodate such a magma chamber, lateral crustal displacement on the order of

1.5 km would have been necessary. Over the 20,000-year formation period proposed by Schmitt et al. (2010), this would imply a deformation rate of ~75 mm/year. However, current geodetic and seismological observations in the Eifel show much lower deformation rates. Seismic moment tensor analyses yield fault displacement rates of only 0.06–1.7 mm/year (Demoulin et al., 2009). Regional uplift rates are similarly modest, averaging ~0.35 mm/year, with localized maxima of ~3.5 mm/year (Garcia-Castellanos et al., 2000; Meyer and Stets, 2002; Mälzer et al., 1983; Campbell et al., 2002), or ~1 mm/year for the broader area

(Kreemer et al., 2020). Given these constraints, no existing tectonic model adequately explains the growth of a magma chamber of the required dimensions at the inferred depth within the estimated timescale. Geophysical investigations around Laacher See have also failed to identify subsurface structures that would support the existence of a remnant magma chamber of ≥11 km³ (magnetotelluric (Ahorner, 1983), gravimetric (Lohr, 1982), seismic (Ochmann, 1988), and magnetic (Pucher, 1992)) The DEKORP project (1994) concluded that no residual magma chamber greater than ~3 km in diameter exists at depths shallower

than 8 km. If residual magma is present, it must exist in a volume that is detectable by geophysical means. However, the estimated caldera collapse volume at Laacher See is only ~0.5 km³ (Viereck and van den Bogaard, 1986). In the absence of a pre-existing volcanic edifice of comparable volume (Schmincke, 2009), this collapse structure should roughly correspond to the erupted volume. The discrepancy—on the order of 6 km³ between the erupted material (plus incorporated country rock) and the observable collapse volume—remains a major unresolved issue in the volcanological assessment of this system.



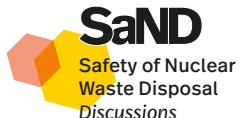

## 5 Definition of the area with a high probability of volcanic activity in the next 1 Ma for the Eifel

A fundamental principle in assessing volcanic hazards in the Eifel region is that risk cannot be evaluated solely on the basis of individual volcanoes. Rather, the entire Quaternary volcanic field must be regarded as potentially active, with both the reactivation of existing volcanoes and the formation of new eruptive centers at previously unknown locations remaining plausible. The Quaternary Rodderberg volcano near Bonn exemplifies that volcanic activity may occur well outside the recognized core areas. Consequently, the region between the West and East Eifel volcanic fields (i.e., the High Eifel) is considered to be part of a unified magmatic system, where the conditions for new volcanic activity persist. A bounding envelope encompassing all known Quaternary volcanic centers can therefore serve as the foundation for hazard zoning and safety buffer modeling. However, this approach does not preclude the possibility of future eruptions beyond the mapped boundary. Thus, any safety buffer for a repository site must account not only for probable eruption zones but also for low-probability, high-consequence events outside the defined envelope. This requires a detailed classification of potential volcanic impacts across various temporal phases of repository operation and long-term containment. For example, a Plinian eruption such as the Laacher See event would pose a severe operational risk even at distances of 55 km, where tephra deposits reached 1 m thickness. Only beyond an additional 10 km buffer would short-term operational disruptions become more manageable. However, given that the operational phase of a repository lasts less than 50 years, the probability of such a high-magnitude eruption occurring within that time frame is negligible. Magma chamber formation requires tens of thousands of years and would be detectable through geophysical and geochemical monitoring well in advance. Consequently, the probability of a major Plinian eruption affecting the operational or post-operational monitoring phase (≤500 years) is considered extremely low. By contrast, for the long-term safety period (1 million years), the likelihood of the formation and eruption of highly evolved magmatic systems increases substantially. In such a scenario, thick tephra deposition—on the order of several tens of meters—could occur. However, from a technical perspective, this is not critical for a deep geological repository, assuming appropriate thermal resistance and mechanical stability of the host rock and engineered barriers. Prior to defining the spatial extent of a potential exclusion zone, several geodynamic considerations must be addressed. Quaternary volcanism in the Eifel is widely attributed to the influence of an actively ascending mantle plume. The future evolution of this plume remains uncertain. Currently, the lithosphere–asthenosphere boundary beneath the Eifel is unusually shallow ($\sim 41 \pm 5$ km), whereas in surrounding regions such as the Upper Rhine Graben, depths increase to $\sim 60 \pm 5$ km (Black Forest, Odenwald) and up to $\sim 78 \pm 5$ km beneath the Swabian Alb and the Vosges (Seiberlich et al., 2013). While long-term relative plate motion might theoretically displace the lithosphere over the plume source, this effect is minimal over the relevant timescales. Gripp and Gordon (2002) estimated a relative plate motion of only $19 \pm 14$ km per 1 Ma. This rate is too slow for significant decoupling of the plume head from its deeper asthenospheric source. Therefore, only mantle anomalies located within the lithosphere are relevant for this assessment. Internal plate deformation, such as the southwestward migration of southwestern Europe west of the Rhine–Rhône line (Tesauro, 2005), is also negligible within the next 1 million years. However, localized tectonic stresses may induce magma ascent on much shorter timescales—from days to weeks—especially in transtensional or strike-slip fault



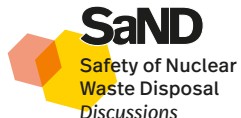

systems. Such structures may also facilitate the development of magma chamber systems in the crust, particularly where they intersect the thermal influence zone of the plume. Even though highly evolved magmas have historically been restricted to the core of the East Eifel, it cannot be excluded that similar systems might develop at the periphery of the Eifel Plume, especially in zones of transtension. While current uplift rates in the Rhenish Massif (~1 mm/year; Kreemer, 2020) may contribute to extensional stress, they are insufficient to generate large magma chambers on relevant timescales. Therefore, uplift alone is not considered a primary driver of evolved melt production. From a conservative risk perspective, the existing bounding envelope of known volcanoes loses importance. Even with an additional 10 km safety buffer, the formation of new volcanic centers outside this zone remains possible due to the broader subsurface extent of the lithospheric mantle anomaly. A cautious approach assumes that volcanic activity could occur anywhere directly above the mantle plume footprint. Two primary scenarios must be evaluated: 1. Mafic (SiO₂-poor) Volcanism: This would likely result in the formation of a maar or scoria cone, with or without lava effusion. If such an eruption occurred at the periphery of the surface plume expression, potential lateral shifts in magma ascent pathways (~15 km; May, 2019) must be considered in addition to a 10 km safety buffer. This yields a minimum exclusion radius of 25 km from the defined plume boundary. 2. Highly Evolved Magmatism: This scenario assumes the development of a magma chamber and a potential Plinian eruption similar to Laacher See volcano. While considered low probability, the consequences would be severe—particularly during the operational phase. Given that chamber formation takes thousands of years and is not currently detectable through geophysical data, this scenario can be excluded for the operational and monitoring phases (≤500 years). For long-term safety assessments, accurate knowledge of the mantle structure and plume dynamics is essential. Several mantle models have been proposed for the Eifel region, yielding differing conclusions regarding the extent and nature of the plume. At present, no single model can be considered definitively accurate. Therefore, a conservative approach must adopt the most extensive interpretation—namely, that of Walker et al. (2005). Accordingly, the reference hazard area is delineated by zones with S-wave velocity anomalies of –1.4% to –6% (represented in red and purple in Figure 2), encompassing the West and East Eifel volcanic fields as well as more peripheral structures such as Rodderberg. The Walker model is also consistent with the region of greatest measured uplift in the Rhenish Shield (Kreemer et al., 2020). By contrast, the 50 km plume margin proposed by May (2019) and adopted by Mertz et al. (2015) based on possible lateral material displacement is not included in the final hazard scenarios, as it lacks sufficient empirical or theoretical support. The processes under discussion may be influenced by statistical uncertainties, tectonic factors, or internal dynamics within the plume itself.

## 6 Map of designated areas in West Germany

The cartographic representation (Figure 4) delineates volcanic hazard zones associated with the Eifel region, classified according to varying degrees of relevance for long-term safety considerations. The innermost perimeter (Line 1) encompasses all known Quaternary volcanic centers and includes a 10 km safety buffer around their outermost occurrences, as stipulated by the German Site Selection Act (StandAG). This zone integrates the West and East Eifel volcanic fields along with the



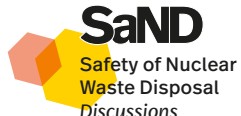

intervening High Eifel into a single coherent hazard area. A dashed boundary (Line 1.1) accounts for the precise distribution of Quaternary volcanoes between West and East Eifel. However, it remains uncertain, as the potential for additional eruption points extending toward Line 1 cannot be excluded. The Line 1.2 extension incorporates the Westerwald region, considering two uncertain age determinations of scattered volcanic deposits. The area enclosed by Line 1 (including Line 1.2) is situated within the plume region as defined by Walker et al. (2005), which extends from 30 to 100 km depth and is demarcated by

dashed Line 2. Most of the ongoing surface uplift (~1 mm per year; Kreemer et al., 2020) occurs within this plume region (dotted Line 3). A 25 km buffer has been applied beyond the outer boundary of the plume (Line 2) in all directions, outlining the maximum extent of potential basaltic volcanism (Line 4). Highly evolved magmatic eruptions are excluded from the plume's peripheral zone over the next 500 years, as well as from the central areas of both Quaternary volcanic fields. The phonolitic volcanoes of East Eifel are located 50 km from the outer margin of the Walker et al. (2005) plume. With an additional

25 km safety buffer, even the highly unlikely scenario of renewed volcanic activity—whether from an existing or newly developing vent—would pose no risk to a repository site outside the safety zone, even during the operational phase. Within this designated boundary, the physical hazard to a repository is considered negligible—not only during operation, monitoring, and retrieval phases but also beyond these timeframes. This safety zone incorporates all relevant hazard factors, while fully covering the uplift zone within German territory. The mantle plume–lithosphere offset due to plate tectonic drift is not relevant

over the next 500 years. Applying these conservative safety margins results in an extension of the hazard zone beyond Germany's western borders into Luxembourg and Belgium. However, these lines are not truncated at the national boundary, even though the assessment area is limited to Germany. A region located north of Vogelsberg and the Westerwald is demarcated by a dashed boundary (Line 5 in Figure 4) as an area of low probability for volcanic activity within the next 1 million years. However, conflicting geophysical evidence from this area precludes a definitive assessment. Older teleseismic

surveys (e.g., Goes et al., 1999; Figure S1, Suppl.) suggest a low eruption probability, which is conservatively incorporated into the hazard evaluation. Although some models propose a mantle anomaly beneath Vogelsberg, the supporting data remain inconclusive. Helium isotope signatures in mofette gases and mineral waters in this region show only slightly elevated mantle contributions, further weakening the case for a deep-seated mantle anomaly

**6.1 Site Hazard Assessment Due to Secondary Effects**

The available age data of the Quaternary Eifel volcanoes indicate a southeastward migration of volcanic activity within both volcanic fields. For the East Eifel, this suggests a potential threat to the Neuwied Basin and nearby reaches of the Middle Rhine Valley. In a worst-case eruption scenario, particularly one involving supercritical water during initial vent opening, slope destabilization and explosive mass wasting in this steep terrain could result in a sudden blockage of the Rhine River. If followed by a lava flow, such a blockage could persist for years. Even more distant lava flows entering the narrow valley from flanking

terrain could generate similar outcomes. In such a scenario, large upstream areas, including portions of the Upper Rhine Graben and adjacent tributaries, would be at risk of extensive inundation. Due to the absence of alternative drainage paths, the lava flow's volume and duration would be critical in determining the onset, extent, and persistence of upstream flooding.

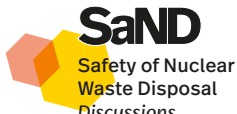

Topographic analysis suggests that a lava-induced barrier could reach over 200 m above sea level. However, for conservative hazard modeling, a maximum dam height of 180 m a.s.l. is assumed, acknowledging the potential for future mitigation
infrastructure. A lava flow of comparable magnitude to that from the Bausenberg volcano could reach the valley floor from adjacent ridges within days to weeks, resulting in immediate backwater effects across large portions of the Upper Rhine Valley. For regulatory safety assessments, such a scenario must be included in site selection evaluations for facilities in the broader Mainz Basin and Upper Rhine Graben (see Figure 09). This applies particularly to the operational phase, as these floods could critically disrupt repository accessibility and above-ground systems.

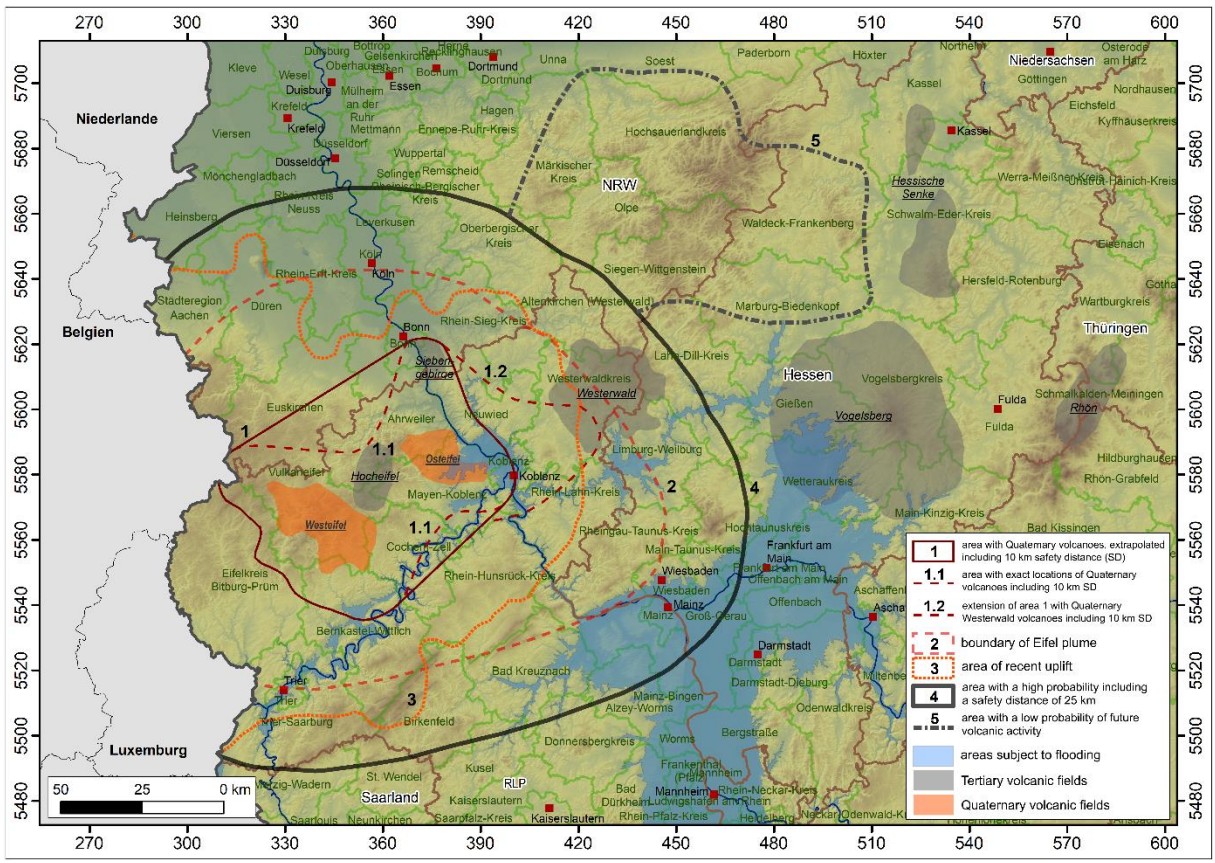


**Figure 4 Region in the Rhenish Massif for which there is a high probability of volcanic activity within the next 1 Ma. Quaternary Eifel volcanism - Line 1: Extrapolated envelope of the West and East Eifel including 10 km safety distance (SD); Line 1.1: Boundary line including 10 km SD with reference to the exact locations of the Quaternary volcanoes in the West and East Eifel field; Line 1.2: Extension of line 1 in relation to possible Quaternary volcanoes in the Westerwald; Line 2: Boundary of the Eifel Plume according**
**to Walker et al. (2005; Figure 2); Line 3: Delimitation of the crustal area with an elevation of 1 mm/a according to Kreemer et al. (2020; Figure 3); Line 4: Boundary line for the area with a high probability of volcanic activity, with 25 km SD starting from line 2. - Dashed line (5): Boundary line for the area with a low probability of future volcanic activity based on mantle anomalies. Blue areas: possible flooding area in case the Middle Rhine Valley becomes dammed by lava flows. Map bases: Volcanological map of the West and High Eifel (Büchel, 1994); Volcanological map of the East Eifel (Bogaard and Schmincke, 1990); Meyer, 1988; Google**
**Earth accessed July 2020; Kreemer et al. (2020) Figure 7a; Walker et al. (2005) here Figure 11; TOP. Basis: SRTM elevation data Germany, national borders Europe: Geoportal of the European Comission (EUROSTAT); County borders loaded from data portal BKG.**


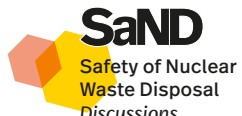

## 7 The Quaternary volcanic region of the Vogtland

In addition to the Quaternary volcanic fields of the Eifel, the Vogtland region constitutes a further area in Germany exhibiting
evidence of recent volcanic activity. However, the number and extent of eruptive centres in the two regions are not comparable.
While over 350 volcanic edifices have been identified in the Eifel, the Vogtland hosts only two alkaline basaltic scoria cones,
situated at the south-western margin of the Eger Basin. In addition, two maars have been confirmed and a further two have
been postulated (Mrlina et al., 2019) (Figure 5). A distinctive characteristic of the Vogtland region is the significant emission
of $CO_2$ from numerous mofettes aligned along fault zones. The $^3He/^4He$ isotopic ratios of these gases indicate a lithospheric
mantle origin (Weinlich et al., 1999), providing compelling evidence for deep-seated magmatic processes. Furthermore, the
Vogtland represents the most seismically active region in Central Europe. Its neotectonic development is associated with
reactivation along major fault systems, some of which correspond to former microplate boundaries within the Variscan
basement (Babuška and Plomerová, 2010; Figure 6).

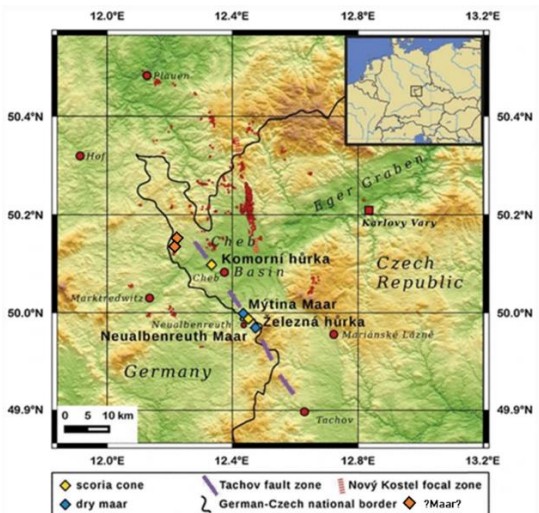

**Figure 5 from Rohrmüller et al. (2018): Locations of the Quaternary volcanoes of the Vogtland (Komorni hurka/Kammerbühl, Zelena hurka/Eisenbühl, Mytina Maar, Neualbenreuth Maar) in relation to the Tachov fault zone. - Swarm quakes (red dots) occur at greate distance from the Tachov fault zone. The two suspected maars at the north-western end (orange squares, Ztraceny rubnik Maar (S) and Baby Maar (N) have been added.**

One of the Quaternary volcanoes is the Kammerbühl (Komorní hůrka), a cinder cone with a small basaltic lava flow near
Franzensbad in the Czech Republic. A second comparable volcano, Eisenbühl (Železná Hůrka), is situated on the Czech side
of the German-Czech border, north of Neualbenreuth (Ulrych et al., 2011). The ages of these volcanoes, determined using four
independent radiometric dating techniques, were reported by Wagner et al. (2002) as 726 ± 59 ka for Kammerbühl and 519 ±
51 ka for Eisenbühl. Rohrmüller et al. (2018) determined the age of the Eisenbühl to be approximately 290 ka. A maar structure
near Mytina was dated to 288 ± 17 ka (Mrlina et al., 2009). Moreover, a crater-like depression near Neualbenreuth has been
interpreted as a possible maar based on geophysical investigations (Rohrmüller et al., 2018). The two confirmed cinder cones,
along with the verified Mytina maar, are aligned along the Tachov Fault Zone, which merges southward into the West



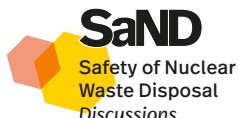

Bohemian Shear Zone. The hypothesised Neualbenreuth maar may represent a southern continuation of this trend. North of Kammerbühl, directly at the German-Czech border, two circular landforms are located approximately 3.5 km north-northwest of Liba. These features may lie along a slightly westward-offset extension of the Tachov Fault. Based on gravity anomalies,

Mrlina et al. (2019) proposed the existence of two additional maars in this area, referred to as the Ztracený Rubník Maar and the Baby Maar (Figure 5).

### 7.1 CO₂ Sources in the Vogtland Region

In several areas of the Vogtland, unusually high volumes of carbon dioxide are released, accompanied by trace amounts of mantle-derived gases. A primary focus of degassing is the Eger Basin, where $CO_2$ fluxes of 28 m³/h at Bublák and 35 m³/h at

Soos have been recorded at the surface. Additional significant emissions occur in Karlovy Vary (Karlsbad), located along the north-western boundary fault of the Eger Graben, and in Mariánské Lázně (Marienbad), situated south of the Eger Rift (Weinlich et al., 1999; Figure S13, Supplement). ³He/⁴He ratios reaching R/Ra values of up to 6 (Bräuer et al., 2018). Bräuer et al. (2011, 2018), based on helium isotope time-series analyses, propose the existence of at least two, likely three, isolated magmatic systems beneath the Moho. These systems correspond to Karlovy Vary, the Cheb Basin, and the Mariánské Lázně

region. Among these, the Cheb Basin currently exhibits the highest level of activity, as indicated by both gas emissions and increased seismicity, including prominent earthquake swarms in 1985/86, 2000, 2008, 2011/12, 2014, 2017, 2018, and 2019. Magmatic activity beneath the eastern Cheb Basin, particularly around the Bublák mofette field, appears to be intensifying. Among the mofette fields associated with the Potočky–Plesná Fault Zone (PPZ; Figure S14, Supplement), Bublák shows the highest $CO_2$ flux, which has significantly increased since 1993, and also exhibits the highest proportions of mantle-derived

helium—both of which are interpreted as indicators of a deep fluid injection zone (Kämpf et al., 2013).

### 7.2 Tectonic framework of the Vogtland and Eger rift

Babuška and Plomerová (2010) attribute the presence of Quaternary volcanism and anomalous $CO_2$ degassing in the Cheb Basin to its unique tectonic setting. The basin is situated at the intersection of three tectonostratigraphic units that collided during the Variscan orogeny: the Saxothuringian and Moldanubian crustal blocks and the Teplá-Barrandian microplate. This

triple junction, located precisely beneath the Eger Basin (Figure S15, Supplement. Between the Saxothuringian and Teplá-Barrandian units, the Eger Rift developed as a well-defined graben. Seismic tomography by Zhu et al. (2012) revealed P- and S-wave travel-time anomalies beneath the Vogtland region, indicating crust–mantle interactions that have resulted in asthenospheric uplift and Moho doming (from ~31 km to ~27 km depth). The lithosphere beneath the Vogtland has been thinned to ~80 km, compared to ~100 km in the Saxothuringian to the north and >140 km in the Moldanubian to the south

(Babuška and Plomerová, 2010; Figure S16, Supplement). Although a mantle plume was initially considered a plausible source for Vogtland magmatism (Schmincke, 2009), teleseismic tomography down to 250 km has not revealed any evidence for a focused upwelling plume (Plomerová et al., 2007). Instead, Wilson and Downes (2006) propose partial melting in the asthenosphere and at the base of the lithosphere, associated with uplift of the asthenosphere to ~80 km depth. Crustal thinning

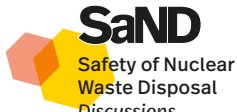

to <28 km has been documented (Geissler et al., 2005; Heuer, 2006; Babuška and Plomerová, 2010; Figure S17, Supplement).
Haase and Renno (2008) support an asthenospheric origin for the Cenozoic magmas of the western Bohemian Massif. According to Babuška and Plomerová (2010), crustal thinning is most pronounced in the western and south-western Cheb Basin, with Moho depths of 28–30 km extending north-eastward. The concentration of volcanic activity along the Tachov Fault Zone correlates with the area of thinnest crust, while major $CO_2$ degassing centres are located slightly to the east, in areas where crustal thickness is still reduced but slightly greater. Only near Marktredwitz, at the western margin of the thinned
crustal domain, has a swarm earthquake occurred that could be linked to enhanced $CO_2$ migration. Hrubcová et al. (2017) postulate a magmatic intrusive complex at the base of the crust in the eastern Eger Basin and south-western Eger Graben, based on seismic, geochemical, and volcanological evidence. Hofmann (2003) had earlier proposed, on the basis of gravity field modelling, a crust–mantle boundary magma reservoir as a more likely interpretation than an asthenospheric bulge (Figure 6). Hrubcová et al. (2017) further suggest that this magmatic body formed by underplating during the late Cenozoic. The
surface projection of the inferred intrusion (Figure S18, Supplement) delineates its zone of influence on gas emissions and volcanism. Interestingly, the Quaternary volcanoes lie just beyond this projection, and the overlap with the zone of minimum crustal thickness is only partial. Figure S19 (Supplement) illustrates a NW–SE lithospheric cross-section through the region of the inferred magmatic intrusion. The highest R/Ra values have been measured in the north-western segment of the section, which is interpreted as the site of a recent magma influx. This active input of mantle-derived material is considered the primary
source of the elevated helium isotope ratios observed in this area. In contrast, the south-eastern portion of the section shows no evidence for an active magmatic supply. Although mantle-derived gases continue to ascend in this area and exhibit elevated R/Ra values, the absolute concentration of mantle helium has remained relatively stable over recent years (Bräuer et al., 2018). This discrepancy suggests a decoupling between gas transport and active magmatism in the south-eastern part of the system, possibly indicating that the mantle source is either waning or no longer actively replenished in this area.

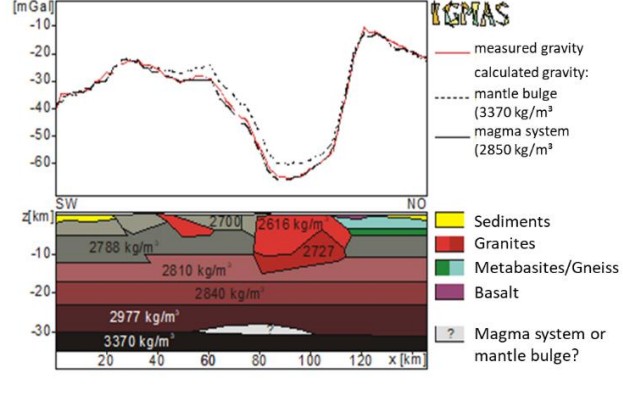


**Figure 06 from Hofmann (2003): Alternative models for the gravitational field. A magma system in the area of the crust-mantle boundary corresponds more strongly to the measured data than a mantle bulge.**

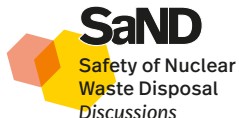

### 7.3 Earthquake swarms

In the Vogtland region, earthquake swarms occur at irregular but recurrent intervals, typically lasting several weeks and
comprising hundreds to thousands of low-magnitude events, rarely exceeding ML 3.5. Unlike typical mainshock–aftershock
sequences, these swarms lack a dominant mainshock (Hainzl and Fischer, 2002). Most of these earthquakes occur at depths
between approximately 7 and 15 km and predominantly take place at the margins of the Eger Basin, where they are associated
with deep-seated faults. The exact mechanisms driving these earthquake swarms remain uncertain. Two leading hypotheses
involve the ascent of magma or supercritical carbon dioxide ($scCO_2$). Depending on the local geothermal gradient, supercritical
water may also contribute (Weinlich, 2014). It remains unclear to what extent volume changes in crustal voids—caused by the
higher mobility of $scCO_2$ compared to liquid water—contribute to seismicity. These ascending fluids are believed to reduce
internal friction along fault planes, promoting stress release predominantly via aseismic creep (Heinicke et al., 2019). Swarm
activity predominantly occurs at the margins of the Eger Basin.

### 8 Determination of the area with a high probability of volcanic activity within the next 1 million years in the Quaternary Vogtland region

Various geophysical studies indicate the presence of mantle anomalies in the Vogtland region, which may account for
Quaternary volcanism. Gases emitted from active mofette fields consistently exhibit elevated ratios of mantle-derived helium
isotopes. Furthermore, earthquake swarms are interpreted in several models as indicative of ascending fluids, likely driven or
influenced by magmatic processes. This interpretation is further supported by geophysical modelling, which suggests the
possible existence of a magma reservoir at or near the crust–mantle boundary. Despite these observations, the tectonic and
magmatic interrelationships remain poorly understood, and no consensus model currently exists that adequately explains the
geological evolution of the Vogtland. Nevertheless, the available data and interpretations permit the development of
preliminary conceptual models that provide a basis for assessing the region's magmatic and tectonic evolution. On the strength
of these models, a high probability of future volcanic activity within the next one million years can be inferred. This assessment
also applies to adjacent areas in north-eastern Bavaria and southern Saxon Vogtland. In contrast, areas further east do not
satisfy the requisite geological and geophysical criteria and are therefore not considered at risk of future volcanism. In
delineating the zone of elevated volcanic hazard, several elements are taken into account. Firstly, the aforementioned
Quaternary volcanoes serve as key reference points. In addition, two recently identified caldera-like depressions near Libá—
which, according to geophysical evidence, are likely to represent maar structures—are included within a conservative hazard
assessment framework. Furthermore, the centres of seismic swarms, along with linearly arranged clusters of earthquakes likely
associated with fault structures, are interpreted as indicators of anomalous gas migration potentially governed by magmatic
influences. These gas migration pathways may serve as future magma ascent conduits under favourable stress conditions.
Accordingly, these seismic centres are treated as potential eruption sites and incorporated into the delineation of the area
deemed to exhibit an elevated probability of future volcanic activity.

SaND
Safety of Nuclear
Waste Disposal
*Discussions*

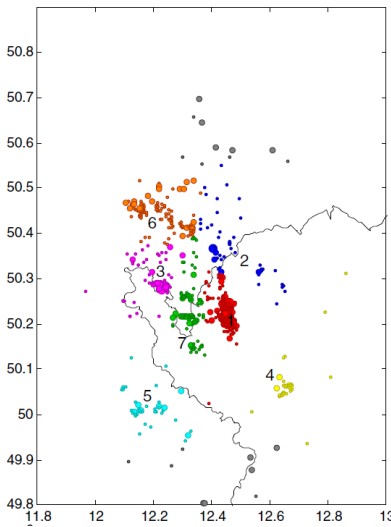

**Figure 7 from Fischer et al. (2014): Epicenters in the Vogtland region in the period 1991–2012. Earthquakes within the respective regions are shown in the same color (1 — Nový Kostel, 2 — Klingenthal, 3 — Kopaniny — Adorf, 4 — Lazy, 5 — Marktredwitz, 6 — Schöneck, 7 — Plesná) according to Horálek et al. (2000). Gray epicenters are not assigned to any of the regions. The size of the dots is proportional to the magnitude. Earthquake swarms after 2012 took place in the Nový Kostel area (1).**

In the area north of Plauen, near Zwickau, Gera, and Leipzig, isolated tectonic earthquakes have occurred, reaching magnitudes of up to 5.0 (Kracke et al., 2000; Leydecker, 2011). As these events are purely of tectonic origin and show no indication of magmatic or fluid-induced processes, they were not considered in delineating the area with elevated volcanic potential. Within Germany, three earthquake swarm centers in proximity to the Eger Basin were included in the assessment: east of Marktredwitz, west of Oelsnitz, and near Klingenthal. The swarm center near Bad Elster lies well within the outlined area, as do the prominent swarm centers of Nový Kostel and Lazy, such that their influence is considered inherently covered due to proximity (Figures 7, 8). Only in Area 2 of Figure 11, a distinct cluster of seismicity southeast of Klingenthal in the Czech Republic was relevant for boundary delineation near Breitenbrunn (northeast of Klingenthal). Although the seismic activity east of Marktredwitz is relatively minor compared to other swarm areas in the region (Dalheim et al., 1997), it was included as part of a conservative hazard assessment strategy. To define the outer boundary of the zone with a high probability of volcanic activity within the next 1 million years, a radius of 25 km from the known Quaternary volcanic centers and earthquake swarm localities was applied. This value includes a 15 km buffer to account for the observed scatter of existing volcanoes and fluid ascent pathways, plus an additional 10 km safety margin, following the methodology used in defining the volcanic hazard zone in the Eifel region.

In the area north of Plauen, near Zwickau, Gera, and Leipzig, isolated tectonic earthquakes have been recorded, reaching magnitudes of up to 5.0 (Kracke et al., 2000; Leydecker, 2011). As these events are purely tectonic in origin and show no evidence of magmatic or fluid-induced processes, they were not taken into account in delineating the area of elevated volcanic potential. Within Germany, three earthquake swarm centres in proximity to the Eger Basin were included in the assessment: east of Marktredwitz, west of Oelsnitz, and near Klingenthal. The swarm centre near Bad Elster lies well within the outlined



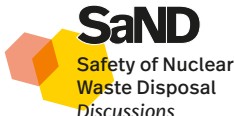

zone, as do the prominent swarm centres of Nový Kostel and Lazy; their influence is therefore regarded as inherently
incorporated due to their proximity (Figures 7, 8). Only in Area 2 of Figure 11 was a distinct cluster of seismicity—located
southeast of Klingenthal in the Czech Republic—relevant for the delineation of the boundary near Breitenbrunn (northeast of
Klingenthal). Although the seismic activity east of Marktredwitz is relatively modest compared to other swarm areas in the
region (Dalheim et al., 1997), it was included as part of a conservative hazard assessment strategy. To define the outer boundary
of the zone with a high probability of volcanic activity within the next one million years, a radius of 25 km was applied around
the known Quaternary volcanic centres and earthquake swarm localities. This distance incorporates a 15 km buffer to account
for the observed dispersion of existing volcanoes and fluid ascent pathways, as well as an additional 10 km safety margin, in
line with the methodology employed in defining the volcanic hazard zone in the Eifel region.

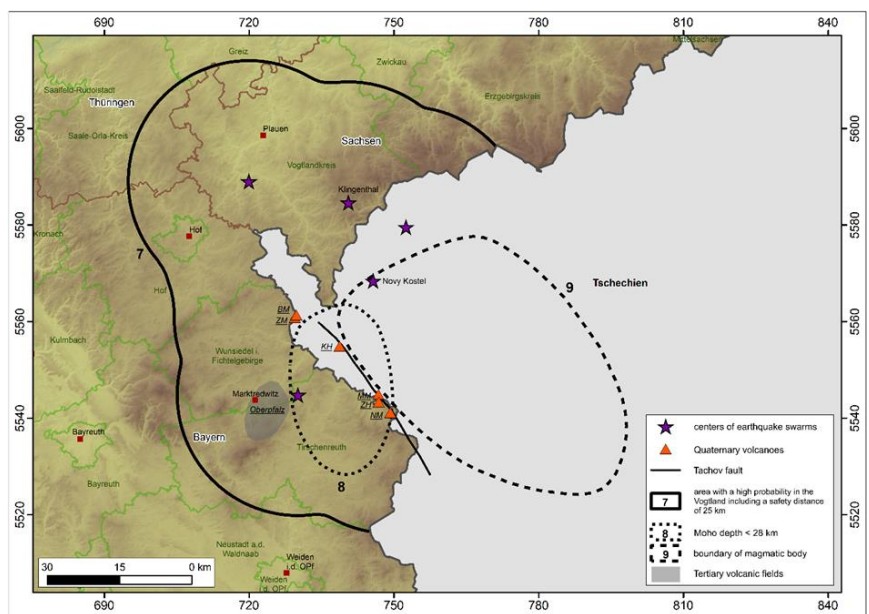

**Figure 08 Map of the Vogtland region for which a high probability of volcanic activity occurring in the next 1 Ma is designated.**
**Reference points are Quaternary volcanoes (orange triangles, including the maar near Mýtina and the two postulated maars near**
**Libá, Ztraceny rubnik Maar (ZM) and Baby Maar (BM); KH, Komorní hůrka; ZH, Železná hůrka; MM, Mytina Maar; NM,**
**Neualbenreuth Maar) and the centers of earthquake swarms in Germany: east of Marktredwitz, southwest of Plauen and**
**Klingenthal, as well as southeast of Klingenthal in the Czech Republic (purple stars). The black line along the volcanoes corresponds**
**to the trend of the Tachov fault. Dashed small ellipse: surface representation of the bulging lithospheric mantle with a Moho depth**
**of less than 28 km according to Babuška and Plomerová (2010); large field: projection of the magmatic body in the lower crust onto**
**the surface according to Hrubcova et al. (2017). Quaternary volcanoes strikingly only occur in the field of the Moho bulge, marginal**
**to the surface projection of the magmatic body. The grey field near Marktredwitz denotes the Tertiary volcanoes of the Upper**
**Palatinate/eastern Upper Franconia. TOP. Basis: SRTM elevation data Germany, national borders Europe: Geoportal of the**
**European Comission (EUROSTAT); County borders loaded from data portal BKG.**

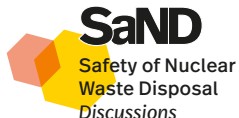

## 9 Current situation in southwestern Germany and delineation of an area with a low probability of future volcanic activity

Teleseismic investigations of the mantle beneath south-western Germany yield contradictory results. While earlier studies point to lithospheric anomalies extending to depths of at least 100 km east of the Black Forest (Goes et al., 1999, Figure S1A, Supplement; Koulakov et al., 2009, Figure S5, Profile 10, Supplement), Zhu et al. (2012) identify a significant mantle anomaly beneath the Urach region only at approximately 275 km depth (Figure 1). This anomaly appears to represent a marginal zone of a broader structure hypothesised to extend northwards beneath the Eifel and Vogelsberg volcanic fields. However, no continuation of this anomaly is detected at depths exceeding 475 km. Given the estimated ascent rates of anomalously hot mantle material—up to 6 cm per year from depths greater than 200 km, as proposed for the Eifel (Ritter, 2005)—the likelihood of future volcanic activity within the next one million years in south-western Germany is considered to be very low. Furthermore, only weak mantle anomalies have been identified beneath the Upper Rhine Graben and the southern Black Forest at depths of 100 km and 75 km, respectively. These anomalies are entirely absent at greater depths in the dataset of Zhu et al. (2012). Gas geochemical measurements from springs in the Urach area, located south of Stuttgart, exhibit helium R/Ra ratios ranging from 0.2 to 0.9 (Ufrecht, 2006). This in itself does not constitute an indication of significant volcanic hazard. Taking into account the advanced age (exceeding 10 million years) of the Urach volcanic field and relying primarily on the seismic results of Zhu et al. (2012), a future volcanic event within the next one million years appears improbable, based solely on the presence of these Tertiary volcanic structures. Nonetheless, some uncertainty persists owing to discrepancies between older teleseismic interpretations and the minor mantle helium signatures observed in spring gases from the region. Furthermore, the recently active Horizontal Albstadt Shear Zone (HASZ), situated south of Stuttgart and Albstadt, may potentially facilitate small-scale magma transport towards the surface. To accommodate these uncertainties, a zone with a low probability of future volcanism is delineated south-east of the Black Forest and south of Stuttgart (Figure 09). The western boundary of this zone follows the extent of mantle anomalies identified by Koulakov et al. (2009; P-wave anomalies at 100 km depth, Figures S5 and S2, Supplement), while the eastern boundary encompasses the extent of the Urach volcanic field.



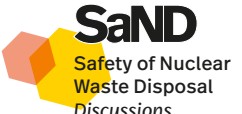


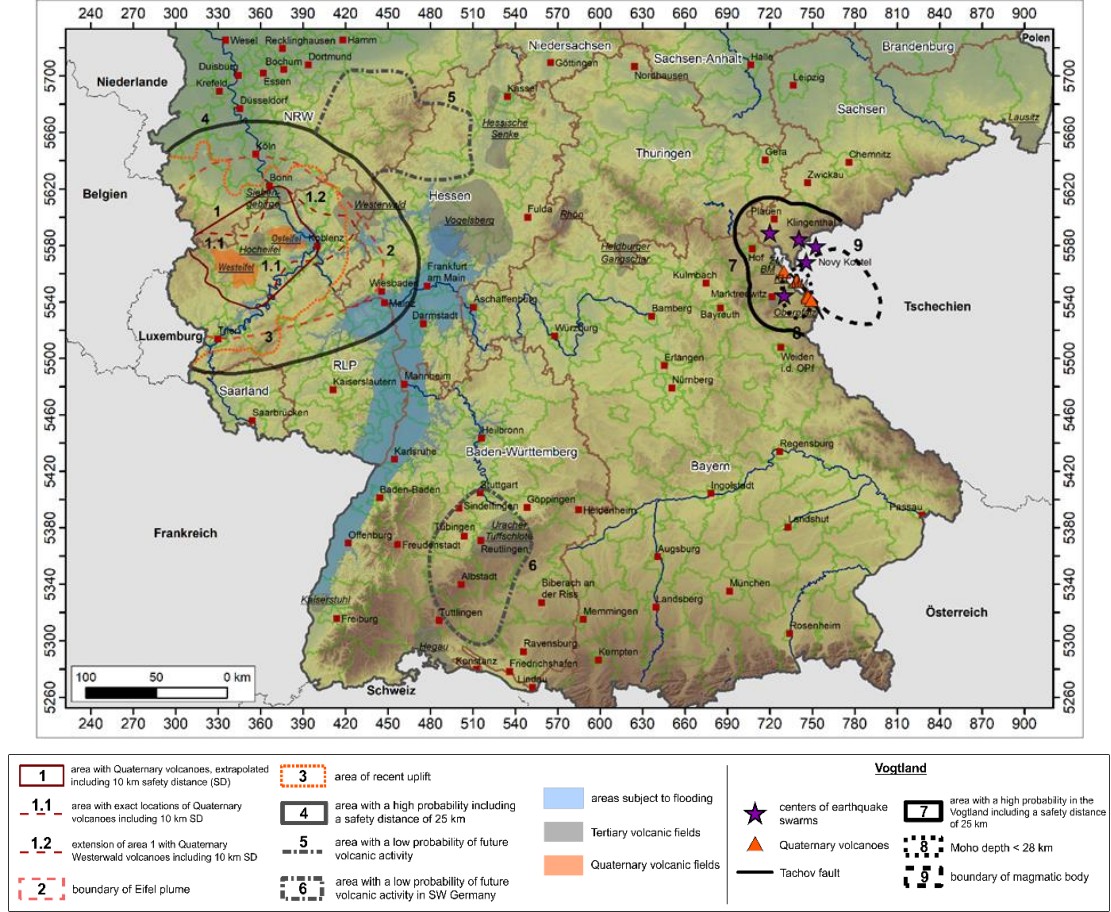

**Figure 09 Overview map of the possible exclusion areas for final disposal of radioactive waste. TOP. Basis: SRTM elevation data Germany, national borders Europe: Geoportal of the European Comission (EUROSTAT); County borders loaded from data portal BKG.**


## 10 Conclusions

In the search for a suitable site for the disposal of high-level radioactive waste in Germany, the Site Selection Act (StandAG) defines a procedure aimed at identifying a repository location that, by statutory requirement, must ensure the highest possible level of safety over a period of one million years. A critical aspect of this evaluation is the potential volcanic hazard associated

with renewed magmatic activity in the vicinity of any candidate site. The present study identifies and evaluates regions within Germany in terms of their susceptibility to volcanic activity over the next one million years—a type of long-term hazard assessment that is unique in the international context. By comparison, the Yucca Mountain Project in the United States (Smith and Keenan, 2005), which had also considered a repository within a geologically young volcanic region, was ultimately abandoned, partly due to concerns over seismic hazard. Although interpretations of the causes of Cenozoic volcanism in

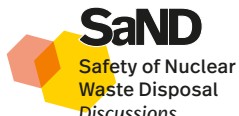

Central Europe vary, there is broad consensus that mantle-derived magmas are produced through localised increases in temperature, decreases in pressure, or fluid-induced melting. Isotopic ratios of gases released from springs and mofettes provide insights into these processes and allow estimates of the mantle contribution. When combined with geophysically derived mantle anomalies and the geochronology of volcanic edifices, these parameters form the basis for delineating zones of elevated volcanic hazard. Within Germany, the Quaternary volcanic fields of the Eifel and Vogtland are considered areas

of high volcanic potential. The Eifel region comprises two subfields—the West and East Eifel—which are assessed as a single unit for hazard purposes. The outermost Quaternary volcanic centres are used as reference points for estimating the spatial extent of possible future activity. In contrast to Vogtland, the lithosphere beneath the Rhenish Massif displays pronounced anomalies in both P- and S-wave travel times, suggesting extensive Quaternary volcanism. The surface projection of these anomalies extends well beyond the current distribution of known volcanoes. Moreover, the region exhibits active surface uplift

of up to 1 mm per year, with a maximum centred in the Eifel, extending beyond the limits of the documented Quaternary volcanic fields. In light of these indicators, a conservative approach has been taken. The boundary of the most expansive plume model (Walker et al., 2005) has been adopted as the primary reference. An additional 15 km margin accounts for the potential activation of a volcanic centre at the periphery of this structure, while a standard 10 km safety buffer is applied to further mitigate uncertainty. The resulting hazard zone encompasses the Quaternary volcanic fields, the inferred plume extent, the

area of maximum uplift, and all springs and mofettes exhibiting significant mantle-derived helium signatures. A possible secondary consequence of an eruption in this region would be the damming of the Rhine River by lava flows or pyroclastic deposits, particularly within its confined valley segments.

The Vogtland region differs markedly from the Eifel, especially in the number and distribution of Quaternary volcanic centres, which represent less than 2% of the total known volcanic features in Germany. In addition, mantle anomalies in the P- and S-

wave fields are weaker and less clearly defined, complicating efforts to identify a mantle plume as the magmatic source. Nevertheless, Vogtland is characterised by elevated seismic activity in the form of recurrent earthquake swarms, interpreted as a consequence of ascending $CO_2$-rich fluids that facilitate stress release at depths of up to 15 km. Mineral springs and mofettes in the area yield helium isotope ratios indicative of mantle-derived input, likely associated with magmatic intrusions in the upper mantle or lower crust. Taken together, these data support the plausibility of future volcanic activity in Vogtland.

The delineation of the hazard zone here follows the same conservative methodology as in the Eifel, with additional consideration given to the centres of earthquake swarm activity. These are interpreted as evidence of significant crustal gas fluxes along faults that may also act as conduits for magma under favourable tectonic conditions. The final boundary maintains an appropriate offset from both the central zone of Moho uplift beneath the Eger Graben—where Quaternary volcanism is confirmed—and from the adjacent eastern sector of the crust, where crust–mantle interactions suggest possible magmatic

intrusion.

Two further regions in Germany have been identified as having a low probability of volcanic activity within the next one million years. The first lies north of the Westerwald and borders the high-probability zone of the Eifel. Its classification rests on earlier teleseismic studies considered in a conservative context, and on slightly elevated mantle helium values observed



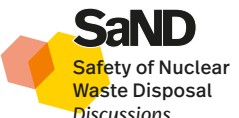

nearby. The second region is situated between Stuttgart and Lake Constance, encompassing the Tertiary Urach volcanic field
and the periphery of the Hegau field. This area is marked by anomalous P- and S-wave signals in the lithosphere and sub-lithospheric mantle, as reported in earlier geophysical studies. While interpretations vary, a conservative assessment assumes elevated mantle temperatures as a possible cause. Slightly increased mantle helium levels in local spring waters, along with the presence of the active Albstadt Shear Zone, suggest that minor magma ascent could theoretically occur in this area. For all remaining Tertiary volcanic fields in Germany, future volcanic activity within the next one million years is not anticipated.
This conclusion is based on their considerable geological age and the absence of supporting evidence from mantle geochemistry, crustal gas emissions, or geophysical anomalies.

**Author contribution**

U.S. prepared the manuscript with contributions from G.J.


**Competing interests**:

The authors declare that they have no conflict of interest.

**Acknowledgements**

Financial support for this project was provided by the BGE (Bundesgesellschaft für Endlagerung), award number **SEVGV3T-19-04-Ol**

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
