# Peer review of "Assessment of possible volcanic hazards in Germany with regard to repository site selection"

_Safety of Nuclear Waste Disposal, 2025_

## Referee Comment (RC3)

**Manuscript by U. Schreiber and G. Jentzsch: Assessment of possible volcanic hazards in Germany with regard to repository selection**

**Reviewers:** T. Dahm (Seismology), H. Kämpf (gas geochemistry)

Being experts in the fields of seismology and gas-geochemistry, we were approached to help in the scientific evaluation of the manuscript after two official reviews were already submitted and several community comments uploaded to the open discussion platform of the journal. While one of the previous reviewers was relatively positive, the other reviewer suggested rejection. Additionally, several shortcomings and errors have been highlighted in the expert reports.

The manuscript, submitted to peer review in SaND, is based on a commissioned report for the Bundesgesellschaft für Endlagerung (BGE), which was prepared by the two authors in 2021 (SEVGV3T-19-04-Ol). According to the authors, this report was taken into account for the designation of potential regions for long-term final storage of radioactive waste. Therefore, the subsequent assessment of parts of the commissioned report in a scientific peer review is delicate, as rejection of the manuscript could discredit the original report. We would like to ensure that our peer review comments are only assessed in terms of scientific quality, innovation and the accuracy of concepts, methods and analyses, and underlying data. Conclusions drawn in the commissioned report may well be correct, even if our and other reviews have critical comments.

The general objective of the manuscript is to record volcanic hazards in Germany over a period of one million years, as a contribution to the question of where in Germany final storage sites for radioactively contaminated waste can be rejected. This is a difficult undertaking, as models and data for recording volcanic hazards within continental plates and distributed volcanic fields are still incomplete. An important aspect of scientific work on this topic is therefore how to deal with the knowledge gaps and how to make quantitative probability-based statements. We therefore focus our assessment on these aspects in particular, regardless of whether the recommended exclusion regions are plausible. One key aspect is whether the submitted work contains scientific innovation and whether the methods, data and prior knowledge used are appropriately up to date. Of course, we also evaluate whether the submitted manuscript meets the usual standards for academic work.

Overall, we conclude that the present manuscript has major issues and cannot be published as it stands. It should be fundamentally revised or rejected. The main reasons are listed below. In addition, we provide comments on specific shortcomings and errors.

**Major comments by T. Dahm:**

The submitted manuscript claims to assess direct and indirect volcanic hazards in Germany over a period of 1 million years and to present them in a hazard map. This is intended to be helpful in the search for suitable sites for the final storage of radioactive waste. The approach is to compile and integrate up-to-date knowledge, input data of seismological imaging, micro-seismicity, geophysical datasets, gas isotope and fluid geochemistry, geological and mineralogical and volcanological state-of-the-art knowledge, and geodynamic modelling. This is a very ambitious undertaking, given that the two authors cover only a small part of the mentioned disciplines and have published comparatively little groundbreaking information on volcanic processes in Germany. This probably explains why there have been so many strong comments from experts on the online platform since the manuscript was submitted. I think the critical community comments are all very good and justified. Often, comments refer to the fact that important work has not been considered, or that the underlying database in this work is not up-to-date and not complete. I feel these are serious concern and needs to be fixed. Additionally, I have comments below. However, my major review will focus

on the methodological approach to the hazard analysis, as this ultimately determines the quality of the submitted manuscript.

In seismology, hazard assessment has been a topic of discussion for many years and is highly developed. Similar for gas-isotope geochemistry. A hazard analysis must not only describe natural events and the database in detail, but is now based on quantitative prediction models and developed statistical models, resulting in probabilistic hazard maps (databases) that quantify not only the probabilities of occurrence themselves but also their uncertainties. The submitted manuscript is of a completely different nature. It takes the approach that future volcanic events are likely to occur where events have already occurred in the past. It is also expected that the possible events could be of a similar type and strength. To account for uncertainties, a safety radius of 10 km or 25 km is proposed, depending on whether we are talking about volcanoes in the Eifel or earthquake swarms in the Vogtland.

As a result, the assessment here provides maps where areas are circled with high or low probability of volcanic hazard. What high and low means is not defined. The approach mostly builds on the past occurrence of volcanic eruptions, and the presence of earthquake swarms today in the Vogtland region (other swarm regions are not discussed). A statistical analysis is missing. An assessment of volcanic ages is missing. Geodynamic models, geodetic surface ground motion rates, and seismological images of upper mantle processes today are discussed, but often not considering newest findings from geodesy and large passive array experiments in the recent years, which are very important as they changed our todays understanding. Also, it appears that upper mantle processes and regional ground deformation models are considered only synoptical and not linked to a probabilistic assessment.

**Specific comments by T. Dahm:**

1.  As mentioned under major comments, the approach to assess the volcanic hazard over a period of one million years is overly simplistic and not up to date. There is also no discussion how the objective is handled in other distributed volcanic fields worldwide. Both needs to be discussed much more.
2.  The various types of input data are crucial for assessing the hazard. The authors review the literature on existing data and models from many different disciplines. This is an enormous undertaking and deserves recognition. Unfortunately, however, this type of review is often short-lived when new findings emerge. This problem is evident in the manuscript and was frequently addressed in the community reviews. I will therefore mention briefly some examples: (1) The statement that there is no geophysical evidence for the size, depth and extend of the magmatic reservoir in the upper crust beneath Laacher See is incorrect (see Zhang et al., 2025). The latest geophysical images indicate that the magmatic reservoir could have been much larger than expected, comprising more than 70 cubic kilometres. (2) The referenced models of shear wave velocity anomalies in the upper mantle are all relatively old and not up to date. For instance, a higher-resolution tomography by Zhu et al. (2015) should be considered. This work has changed our understanding of upper mantle anomalies. I used this 3D S-wave velocity model to quickly check the statements of the submitted manuscript here. My comparison is very different (see Figure 1 below). The example plotted in this quick analysis clearly shows that, in addition to the Eifel and Vogtland regions, there are other anomalous zones in the upper mantle beneath Germany (e.g. in the Harz region) that have not been taken into consideration in the assessment here. Today, there are even better shear wave anomaly maps published by different groups. These need to be taken into consideration. The reason why recent 3D velocity models are superior to work published 20 or 30 years ago is very simple – the seismic networks and stations densities improved significantly in the last 10 to 5 years. This is a major factor for resolving

velocity anomalies in the upper mantle and crust. (3) The interpretation of the Ochtendung seismicity zone as a system of tectonic, en-echelon strike slip faults is weird, as it refers to unpublished hypothetical models by one of the authors which is not based on data and not established. For example, the analysis of a large-N nodal seismic array in the East Eifel, consisting of 500 stations with interstation distances of 1-2 km, resolved and interpreted the Ochtendung seismicity much better than before (e.g. Laumann et al., 2025). This study ind

3. icates that the seismicity in the Ochtendung zone is controlled by ongoing fluid flux from depth (recharge?) and has a non-tectonic origin.

4. Mantle derived CO2 and the associated helium isotope ratios (Ra values) are reported as one important indicator for ongoing magmatic activity at depth. I agree with this in general. However, the authors do neither present and discuss a comprehensive review of the current work and knowledge on this (e.g. analysis by May, Bräuer, Bekaert, Woith, Shaw is missing - the authors mostly cite themselves with relatively old work). They also do not explain why and how Ra values should help a probabilistic assessment of volcanic hazards, or what exactly can be derived from gas geochemistry and isotopy studies. For instance, continuous fluid monitoring networks and renewed campaigns have been deployed in recent years in the Eifel and the Eger region (not referenced). From these data, local transients have been reported, for instance in a significant increase in Ra values in the Laacher See region. What does it mean for a probabilistic assessment. How do we know that our measuements are complete (for instance, do we have fluid measurements from the Harz region?), These questions should be addressed.

5. I wonder why there is no table or database provided on ages of all volcanoes in Germany. Isn't this an essential part of any volcanic hazard assessment. For instance, the Boosener maar, which is one of the youngest maars in the Eifel, and not located in either the East and West Eifel, is not discussed. The Boosener maar was affected in October 2025 by a shallow earthquake of magnitude M>2. Reports indicate that 14 wild boars could have been killed by CO2 at the time of the earthquake. How does the Boosener maar and its unusual location compile with the volcanic hazard model in the manuscript?

6. High resolution seismic tomography in the East Eifel resolve a strong velocity anomaly beneath the Laacher See and Neuwied basin much larger than anticipated (Zhang et al., 2025). The estimated DRE of 6.3 km3 of erupted material 13 ka ago is only a small portion of this velocity anomaly. The authors question against reviewer 1 whether the eruption 13 ka ago could have been smaller than estimated in previous published work. With regard to the newest finding from seismology, the questions to be asked is whether a possible future eruption over a period of 1 million years could be larger that the VEI 6 eruption 13 ka years ago. Seismology tells us that the structural setting for such a scenario possibly exist.

*Other comments by T. Dahm:*

Line 68. The concept of safety buffers is introduced to accommodate low-probability eruption scenarios that produce tephra deposits. However, probabilities of different eruption scenarios are not defined in the later chapters, and safety buffers are later defined ad hoc without further analysis.

Line 78: A lithosphere-crust boundary is not defined. Note that the crust is part of the lithospheric plate.

Line 90 – amnd lines 195-205): The statement, that the upper mantle is "normal" outside the Eifel anomaly (e.g. at 70 km depth), is wrong. Reduced shear wave velocities are found beneath the Eifel, beneath the Massif Central, the Pannonian basin and beneath the Harz region (see figure below). Novel continuous GNSS data also indicate uplift in the Harz and parts of Brandenburg, which is not discussed in the assessment here. For example, when forecasting possible developments over one million years, why can the Harz region be exluded?

Line 74: The reference on seismological velocity tomographies do not include recent work with significant results for Central Europe. Beside Zhu et al. (2015) the lates work from the Kiel group should be considered.

Line 108: The statement, that a direct connection between velocity anomalies at depth of about 470-620 km in the upper mantle do not exist, is on contrast to recent work and new modelling results, e.g. Li et al., 2025. Latest geodynamic modeling is relevant for the volcanic hazard assessment.

Line 220 ff. Current uplift rates in the Eifel region can alternatively be explained by processes of magmatic underplating and Moho-depth intrusions (Silveri et al., 2024).

Line 237: I don't understand why the absence of phonolitic rocks at the surface indicate that the uplift is a very recent phenomena. And what follows then – do we expect major phonolitic eruptions in future in the West Eifel?

Line 248: The stress field in the Eifel and in the Rhine graben experienced since Tertiary a systematic anti-clockwise rotation of the maximal horizontal stress. Such a systematic trend influenced the tectonic evolution and the seismic hazard till today. It is expected, that also the volcanic hazard is influenced by a systematic stress rotation. The authors do not consider and discuss this effect and its influence on the volcanic hazard. Additionally, we see today a spatial variation of the stress field across the Neuwied basin. Whether this spatial variation is related to uplift or directly to magmatic processes or other reasons is not solved. It should be discussed.

(Line 260) Quaternary volcanism in the Westerwald is referenced, but not further discussed. This is justified because the age determination were uncertain. It is important for a proper assessment as here to identify knowledge gaps and what their influence on the probability of future volcanic eruptions is.

Line 250-260 and line 274: The fact that one of the young volcanic centres are located in the Hocheifel Tertiary field, e.g., the Boosener Maar, are completely discarded in the discussion.

Chapter 4.4, lines 297-303: The discussion on supercritical fluids like CO and $H_2O$ falls short and is not sufficient. No phase diagram is given. For instance, $CO_2$ is in a supercritical state until about 1 km below the surface. Water is not in a supercritical state, except it would be extremely hot (e.g. Büyükakpinar et al. 2025). I would have expected a broader view that includes knowledge in eruption mechanism from worldwide

Line 314: Citation on the most recent dating of the Laacher See eruption is missing

Line 323: A magma chamber evolution is estimated to take several tens of thousands of years. The given references to Annen appears not appropriate to me, as this consideres melt and reservoir formation processes in the lower crust. I would expect a better discussion from examples worldwide, including the question how long such a reservoir can stay in a critical state if occasionally replenished by influx of fresh magma. From what I know, the time of emplacement and the initial formation of a magma reservoir must have occurred much earlier than anticipated in the manuscript, maybe even started 700 ka ago.

Lines 365 – 370: The authors state that a major Plinian eruption will not be critical for a nuclear waste repository during the initial and operational phase. However, the arguments provided are qualitative and not further detailed.

Line 381: Authors conclude that the extensional stresses caused by current widespread uplift are insufficient to generating large magma reservoirs over relevant timescales (what is relevant – not defined). However, this has not been analysed in the study, references are not provided. Statement should therefore be deleted.

Line 385: The authors introduce a qualitative assessment that volcanic activity can occur anywhere above the so-called mantle plume. However, anomaleous mantle velocity and uplift is also indicated beneath Harz and parts of Brandeburg. Why are these regions discarded? .

Line 397 ff. The reference hazard zone is delineated by a zone of S-wave anomalies of -1.4% to 6%, according to Walker et al. 2005. Why -1.4 - -6%? Why only based on one paper that is relatively old, when better S wave anomaly images exist today?

Line 401: The plume margin by May (2019) is excluded in the assessment of this paper (May 2019, is not properly cited, why). Additionally, a published paper pre-dating the work by May, i.e. Mertz et al., 2015, is discarded with the argument it adopted the May (2019) report which was published 4 years later. From all I know, the May report is a corner stone for the assessment of the nuclear waste repositories in Germany. It considered the expertise of a broad group of established experts from a broad range of disciplines.

Line 408: How to justify the 10 km safety buffer as standard around known volcanic edifices in the Eifel, while the 25 km safety buffer is drawn around earthquake swarms in the Vogtland region? Earthquakes swarms have occurred in other regions in Germany (e.g. Meerseburg in Rineland Palatine, 2023-2025). Why are these discarded. An assessment of historical swarms in Germany is not provided.

Line 423: A physical hazard is considered negligible for a repository in 25 km distance from a newly formed eruption? Neither a probability is given nor it is explained how the probability is estimated to consider it negligible.

Line 424: It is mentioned that the 10 km safety zone considers all relevant hazard factors. However, hazard factors have never been specified in a quantitative manner!

Line 427 ff: The region north of Vogelsberg and Westerwald is considered of low probability for an eruption during the next million years. However, a probability is not given and not estimated. The exclusion is justified by inconclusive data, i.e. lack of knowledge. This is by itself contradicting the previously introduced approach to be conservative.

Figure 4, 8 and 9 present the major results, i.e. the "volcanic multi-hazard maps". They are all very difficult to read. Contentwise, no probabilities are provided and additionally optional hazard areas are indicated, where it remains unclear what they mean. Overall, the result figures should be improved. For instance, the manuscript claims that faults could control magma ascent through the crust. The hazard map do not show major faults or graben systems (only one poorly known fault is indicated for Vogtland).

*Chapter 7-8: Vogtland and Cheb basin regional*

... there are more comments and concerns on the chapter of Vogtland and the Cheb basin – not included this first review, as the list is already long. If the editor decides to save the manuscript, this part can be complemented.

The Quaternary Boosener Maar located in the Hocheifel questions whether Tertiary field may have a potential to be re-activated over a period of one milloon years. This could affect regions like the Rhön, Urach, Siebengebierge, Hegau, and more). This aspect should be discussed in more depth.

Line 640: the "Horizontal Albstadt Shear Zone in South Germany is mentioned and indicated as a potential zone of future volcanism. No reference is given. Close to Albstatt earthquakes have been recorded, including M>6 events in the last century. However, these events are to my knowledge associated with tectonic and not with magmatic processes. The authors identify regions SE of the Black forest and the Albstatt zone to be potentially affected by volcanism, without detailed explanations.

**Specific comments by H. Kämpf:**

My own field of expertise is focused in the research of fluids (mostly gas) of basement rock areas in geodynamic active regions of Central and Western Europe. I performed since 1991 isotope geochemical research studies (C, He, N) and gas flux measurements in the Vogtland/Western Bohema area (regional mappings and time series researches). In 2001 I started study of the free gas phase of mineral water and mofette of the Eifel area as comparative study to the western Eger Rift.

I primary concentrate my comments on sections 3, 7 and 10.

*Section 3. Isotopy of mofette gases in Germany and adjacent areas*

This section has to be reconstructed and reworked because of (1) line 129: update of Table S1 (Supplement), (2) Consideration of gas isotope results from S-Eifel (Heckenmünster) and (3) line 133-135: Consideration of noble gas isotope results from western Eger Rift area.

(1) Table S1 (supplement) has to include results from Bräuer et al. (2013, Tab. 1a, b: loc. nbr: 3-5, 8-10, 17, 19, 24, 26, 27).

(2) Consideration of gas isotope results from the S-Eifel (Heckenmünster).
Because Schreiber and Jentzsch don´t present and discuss stable isotope results from S-Eifel I shortly summarize and discuss here this aspect:
- According to Bräuer et al. (2013), Bekaert et al. (2019, 2020) and Marty et al. (2020) the gas from mofettes in the Southern Eifel near Heckenmünster show in the Ar-Kr-Xe isotope ratio MORB origin (Upper mantle, asthenosphere)! This is in confirmation of the Ne-Ar-N isotope results friom Bräuer et al. (2013), while the He-isotope ratio tend to subcontinental lithospheric mantle (SCLM) origin.
- However, at the area of the Southern Eifel area is no or no known volcano of Quaternary or Neogene age.
- I propose to consider published results from degassing field at Heckenmünster in the high probability volcanic activity analysis and in Fig. 4.
- One further aspect of hazard analysis may the splitting distance between the Mosel-river valley and the mofette field. A possible secondary consequence of an Eruption in the South Eifel would be a damming of Mosel River by lava flows or pyroclastic deposits. Because at the Luxembourg/France border is working the French nuclear power station of Cattenom, based on 4 pressurized water reactors, this may also to consider in the risk-analysis.

(3) Consideration of noble gas isotope results from western Eger Rift area.
- Line 133/134: Cite please beside Bräuer et al. (2018) also Daskalopoulou et al. (2025).
- According to Daskalopoulou et al. (2025) the Ne isotopic ratios in the free fluid phase of mofettes suggest a more nucleogenic/crustal component with respect to MORB compositions (Upper mantle, asthenosphere) while the He-isotope ratio tend to lithospheric origin (SCLM).

Summing up melt infiltration from asthenospheric depths into the lithosphere beneath the Eifel and the western Eger Rift/Cheb Basin was scientifically proven by gas-isotope research. The discrepancy   between Ar-Kr-Xe and Ne-Ar-N isotope ratio tend to MORB and He isotope ratio trend to SCLM may cause by subduction-related crustal contamination of He during the Variscan Orogeny. In situ radioactive decay of U and Th-rich crustal material likely produced $^4$He and lowering the $^3$He/$^4$He ratios with respect to MORB (Daskalopoulou et al., 2025).

*Section 7: The Quaternary volcanic region of the Vogtland (change Vogtland by: Western Eger Rift/CZ-D)*

- Line 483: "Mrlina et al. (2007) determined the age of the Eisenbühl…" please change Rohrmüller et al. (2018) by Mrlina et al. (2007).
- Proposal for new sentence: Combined geoelectric, volcanologic and mineral magnetic investigations of tephra layers at surroundings of the Mytina maar gave first insights to the hazard potential of a Quaternary maar eruption at the western Eger Rift (fall deposits, flow deposits), Flechsig et al. (2015), Lied et al. (2020).

*Sections 7.1: CO2 Sources in the Vogtland Region and 7.2: Tectonic framework of the Vogtland and Eger rift*

- Line 497: Proposal for new sentence: The $CO_2$ soil degassing of the of the Hartousov and Bublak mofette fields show that PPZ act as fluid cannels to depth (CO2 conduits), Kämpf et al. (2019).
- Line 516: Replace Schmincke (2009) by Granet et al. (1995).
- Line 534: Proposal for new sentence: Bräuer et al. (2009) detected in 2006 a progressive spatial and temporal increase of mantle-derived helium in escaping gases at surface reached the SCLM range and points to fluid injection channels reaching down to the lithospheric mantle. This fact strongly indicate at least two upper mantle-derived magma intrusions in 2000 and 2006 (Bräuer et al. 2018).

*Section 10: Conclusions*

- Line 665: Proposal for new sentence: The Eifel region comprises two subfields marked by Quaternary volcanism and CO2 degassing of magmatic origin and a small newly detected mofette field in the Southern Eifel without known volcanism. The volcanic and degassing centers are used as …
- Line 675: Proposal for new sentence: A possible secondary consequence of an Eruption in the East Eifel and the South Eifel would be a damming of Rhine River and Mosel River by lava flows or pyroclastic deposits …

**Supplementary information to our review comments, including references:**

[Figure]

Figure 1: Shear wave velocity anomalies in the upper mantle in 70 km depth beneath Central Europe. Plotted from the database provided by Zhu et al., 2015, GJI. Ceneozoic volcanic field are indicated by grey pologones (Massif Central, Eifel, V=Vogelsberg, R=Rhön, EG=Eger graben, K=Kaiserstuhl, etc.) Yellow diamonds indicate regions of mantle-derived CO2 emissions.

References:

Zhang, H., Dahm, T., Haberland, C., Isken, M. P., Laumann, P., & Büyükakpınar, P. (2025). The upper crustal structure of the Eifel volcanic region (southwest Germany) from local earthquake tomography using large-N seismic network data. *Journal of Geophysical Research: Solid Earth*, 130, e2025JB031338. https://doi.org/10.1029/2025JB031338

P Laumann, T Dahm, G Petersen, P Buyukakpinar, H Zhang, M Isken, B Schmidt, Microseismicity Reveals Fault Activation and Fluid Processes Beneath the Neuwied Basin and Laacher See Volcano, East Eifel, Germany, *Geophysical Journal International*, 2025;, ggaf475, https://doi.org/10.1093/gji/ggaf475

Hejun Zhu, Ebru Bozdağ, Jeroen Tromp, Seismic structure of the European upper mantle based on adjoint tomography, *Geophysical Journal International*, Volume 201, Issue 1, April 2015, Pages 18–52, https://doi.org/10.1093/gji/ggu492

Büyükakpınar, P., Dahm, T., Hainzl, S., Isken, M., Ohrnberger, M., Doubravová, J., Wendt, S., & Funke, S. (2025). Modelling of earthquake swarms suggests magmatic fluids in the upper crust beneath the Eger Rift. *Communications Earth & Environment*. https://doi.org/10.1038/s43247-025-03019-0

Yingying Li, Bernhard Steinberger, Sascha Brune, et al. Intraplate volcanism driven by slab-plume interaction: Numerical modeling and its application to the Eifel, Massif Central and Hainan volcanic areas. *ESS Open Archive* . August 29, 2025

Silverii, F., Mantiloni, L., Rivalta, E., & Dahm, T. (2023). Lithospheric sill intrusions and present-day ground deformation at Rhenish Massif, Central Europe. *Geophysical Research Letters*, 50, e2023GL105824. https://doi.org/10.1029/2023GL105824

References specific for gas-geochemistry:

Bekaert, D.V., Broadley, M.W., Caracausi, A., Marty, B. (2019). Novel insights into degassing history of Earth´s mantle from high precision noble gas analysis of magmatic gas. EPSL 525, 118886, 1-13. doi:10.1016/j.epsl.2024.118886.

Bekaert, D.V., Caracausi, A., Marty, B., Byrne, D.J., Broadley, M.W., Caro, G., Barry, P.H., Seltzer, A.M. (2020). The low primordial heavy noble gas and 244Pu-derived Xe contents of Earth´s convecting mantle. EPSL 642, 115766, 1-12. doi:10.1016/j.epsl.2019.115766.

Bräuer, K., Kämpf, H., Strauch, G., 2009. Earthquake swarms in non-volcanic regions: what fluids have to say. Geophys. Res. Lett. 36, L17309. http://dx.doi.org/10.1029/2009GL039615.

Daskalopoulou, K., Niedermann, S., Wilke, F.D.H., Zimmer, M., Woith, H., Glodny, J., Geissler, W.H., Kämpf, H. (2025). Characterisation of deep intra-continental magma reservoirs – Insights from noble gases and p-T estimates into the western Eger Rift (Czech Republic). Chemical Geology, 681, 122722. Doi: 10.1016/j.chemgeo.2025.122722.

Flechsig C, Heinicke J, Mrlina J et al (2015) Integrated geophysical and geological methods to investigate the inner and outer structures of the Quaternary Mýtina maar (W-Bohemia, Czech Republic). Int J Earth Sci 1:1. doi.org/10.1007/s0053 1-014-1136-0.

Granet , M., Wilson, M., Achauer, U. (1994). Imaging a mantle plume beneath the French Massif Central, EPSL, 136, 28 l - 296.

Hrubcova, P., Geissler, W.H., Bräuer, K., Vavrycuk, V., Tomek, C., Kämpf, H. (2017). Active magmatic underplating in western Eger Rift, Central Europe. Tectonics 36.

Kämpf, H., Broge, A.S., Marzban, P., Allahbakhshi, M., Nickschick, T. (2019). Nonvolcanic carbon dioxide emission at continental rifts: the Bubl´ak mofette area, western Eger rift, Czech Republic. Geofluids 2019, 4852706.

Lied, P., Kontny, A., Nowaczyk, N., Mrlina, J., and Kämpf, H. (2020). Cooling rates of pyroclastic deposits inferred from mineral magnetic investigations: a case study from the Pleistocene Mýtina Maar (Czech Republic), Int. J. Earth Sci., 109, 1707–1725, doi.org/10.1007/s00531-020-01865-1.

Marty, B., Almayrac, Barry, P.H., Bekaert, D.V., Broadley, W., Bryne, D.J., Ballentine, C.J., Caracausi, A. (2020). An evaluation of the C/N ratio of the mantle from natural CO2-rich gas alalysis: geochemical and cosmochemical implications. EPSL, 551, 116574, 1-12. doi:10.1016/j.epsl.2020.116574.

Moreira, M., Pouchon, V., Muller, E., Noirez, S. (2018), The xenon isotopic signature of the mantle beneath Massif Central. Geochem. Persp. Lett., 6, 28-32. doi: 10.7185/geochemlet.1805.

Mrlina, J., Kämpf, H., Geissler, W.H., van den Bogaard, P. (2007). Assumed Quaternary maar structure at the Czech/German borderbetween Mýtina and Neualbenreuth (western Eger Rift, Central Europe): geophysical, petrochemical and geochronological indications. Z. geol. Wiss. 35, 4–5, 213–230.